# The successor representation subserves hierarchical abstraction for goal-directed behavior

Sven Wientjes[ID]*, Clay B. Holroyd

Department of Experimental Psychology, Ghent University, Ghent, Belgium

* wientjes.s@gmail.com

## Abstract

Humans have the ability to craft abstract, temporally extended and hierarchically organized plans. For instance, when considering how to make spaghetti for dinner, we typically concern ourselves with useful "subgoals" in the task, such as cutting onions, boiling pasta, and cooking a sauce, rather than particulars such as how many cuts to make to the onion, or exactly which muscles to contract. A core question is how such decomposition of a more abstract task into logical subtasks happens in the first place. Previous research has shown that humans are sensitive to a form of higher-order statistical learning named "community structure". Community structure is a common feature of abstract tasks characterized by a logical ordering of subtasks. This structure can be captured by a model where humans learn predictions of upcoming events multiple steps into the future, discounting predictions of events further away in time. One such model is the "successor representation", which has been argued to be useful for hierarchical abstraction. As of yet, no study has convincingly shown that this hierarchical abstraction can be put to use for goal-directed behavior. Here, we investigate whether participants utilize learned community structure to craft hierarchically informed action plans for goal-directed behavior. Participants were asked to search for paintings in a virtual museum, where the paintings were grouped together in "wings" representing community structure in the museum. We find that participants' choices accord with the hierarchical structure of the museum and that their response times are best predicted by a successor representation. The degree to which the response times reflect the community structure of the museum correlates with several measures of performance, including the ability to craft temporally abstract action plans. These results suggest that successor representation learning subserves hierarchical abstractions relevant for goal-directed behavior.

**Data Availability Statement:** All data and code used for statistical analyses and online task administration are available on OSF (https://osf.io/n2jcz/).

## Author summary

Humans have the ability to achieve a diverse range of goals in a highly complex world. Classic theories of decision making focus on simple tasks involving single goals. In the current study, we test a recent theoretical proposal that aims to address the flexibility of

**Funding:** The current work was supported by grant 787307 from the European Research Council (ERC) awarded to CBH under the EU's Horizon 2020 Research and Innovation programme. SW was also supported by grant 11E3823N from Research Foundation Flanders. The funders had no role in study design, data collection and analysis, decision to publish, or preparation of the manuscript. ERC Horizon 2020: https://research-and-innovation.ec.europa.eu/funding/funding-opportunities/funding-programmes-and-open-calls/horizon-2020_en Research Foundation Flanders: https://www.fwo.be/en/.

**Competing interests:** The authors have declared that no competing interests exist.

human decision making. By learning to predict the upcoming events, humans can acquire a 'model' of the world which they can then leverage to plan their behavior. However, given the complexity of the world, planning directly over all possible events can be overwhelming. We show that, by leveraging this predictive model, humans group similar events together into simpler "hierarchical" representations, which makes planning over these hierarchical representations markedly more efficient. Interestingly, humans seem to learn and remember both the complex predictive model and the simplified hierarchical model, using them for distinct purposes.

## Introduction

Classically, research on decision making in humans and animals has focused on comparing choices with differently valued outcomes occurring in the immediate future [1,2]. This problem can be formalized in the framework of reinforcement learning (RL; 3) and has led to a rich understanding of the relationship between learning and dopaminergic responses in the midbrain [4]. However, naturalistic decision making often involves long-range dependencies between multiple choices and temporally distal outcomes [5]. Recently, the interest in biological mechanisms subserving such temporally extended decision making has surged [5–8]. Temporally extended behaviors are problematic for standard RL models, which are highly sensitive to the combinatorial explosion of possible states and actions available in real world sequential decisions [9,10]. One approach to alleviate this combinatorial explosion is to generalize knowledge over states or actions, which is called hierarchical reinforcement learning [6,11]. This involves the learning of effective hierarchical representations that allow for meaningful generalization, but the neurocomputational mechanisms underlying this remain to be determined [12].

Previous accounts of hierarchical representation learning have mainly relied on graph-theoretic analysis of the structure of the task [13–16]. However, this requires a fully accurate representation of this task structure to begin with, which in reality often has to be learned during the task itself [17]. The successor representation (SR) [18] is a reinforcement learning algorithm specifically concerned with learning task structure based on a biologically plausible learning rule [19–21]. Notably, it has been suggested that the successor representation can be leveraged to discover hierarchical representations [22,23]. The successor representation learns a multi-step prediction of the task structure, which allows for fast and flexible behavior in the face of changing goals [24]. Representing the task structure in this way balances fast computation commonly associated with so-called model-free reinforcement learning against the flexibility commonly associated with so-called model-based algorithms which require more computational time to adjust behavior flexibly [25]. Interestingly, it has been shown that the behaviors of both humans and rats exhibit some of the limitations in decision making specifically associated with the successor representation [26,27], which can capture some human and rat learning effects better than typical model-free or model-based RL algorithms do [28]. Additionally, the successor representation can explain how people segment events based on the graph topology of the events, capturing higher-order structure in the environment [29–31]. However, to our knowledge no one has yet investigated whether the successor representation is leveraged for the purpose of extracting hierarchies relevant for goal-directed behavior.

In the current study, we investigate whether the successor representation provides a basis for hierarchical abstraction that is actually used for planning goal-directed behavior. We designed a task that afforded a simplified choice policy based on community relationships observable in the lower-order task structure. Participants played a computer game wherein

they acted as a tour guide in a museum, leading visitors to rooms with specific paintings that they requested to view. A new request was made every time a painting was located, which encouraged the participants to learn an internal model of the museum in order to find the paintings as efficiently as possible. Unbeknownst to the participants, the museum consisted of three different wings, with each wing containing five highly interconnected rooms. The participants made binary choices to navigate between the rooms of the museum, where each choice was probabilistically mapped to two of the four possible neighboring rooms. These action-outcome mappings afforded a simple policy to move between wings, as consistently selecting one key yielded on average shorter paths to one of the two remaining wings, whereas the other key yielded shorter paths to the other wing.

This task design allowed us to test two key predictions. First, we predicted that participants would learn an internal model of upcoming rooms while they navigated through the museum [23]. We predicted that aspects of this internal model would be reflected in response times, with more expected rooms yielding faster response times [29], and expectations and prediction errors about reward also influencing response times [32–36]. Second, we predicted that participants would extract and utilize hierarchical information about the museum, which would be reflected in their choice behaviors. In particular, we expected that their choices would be sensitive to the higher-order structure of the museum wings rather than only to the lower-order structure of the museum rooms, which would enable faster planning using less cognitive effort [6,37].

Following a series of preregistered analyses, we tested the nature of the learned internal model separately for the response time and the choice data. We compared several candidate models: An optimal "model-based" model that perfectly represented the lower-order structure of the rooms of the museum, an "explicit hierarchical" model that perfectly represented the higher-order structure of the wings of the museum, and a "successor representation" model that learned multistep predictive representations of the rooms sensitive to higher-order community structure of the wings. We found that the successor representation model accounted for the response times the best, whereas the explicit hierarchical model accounted for the choice behavior the best. In an exploratory follow-up analysis, we examined the "discount factor" parameter of the successor representation, which arbitrates the degree to which the community structure is reflected in the learned predictions. We observed that individual differences in the fit of the discount factor correlated with 1) effective hierarchical abstraction as reflected in choice behavior, 2) accumulated reward compensation at the task, and 3) performance on a secondary task where the participants were asked to reconstruct the rooms of the museum from memory. In all cases, discount factors that reflected the higher-order structure of the wings more strongly predicted better performance, including a more accurate reconstruction of the lower-order structure of the rooms of the museum in the secondary task. These results support the hypothesis that the successor representation subserves effective hierarchical abstraction for goal-directed behavior.

## Results

### Experimental design

We initially ran a pilot study and preregistered a sample and analysis plan for the current study (https://osf.io/n2jcz/). Based on the preregistration, we collected a total number of 141 participants on the online platform Prolific [38]. We excluded 21 participants based on preregistered exclusion criteria that ensured that the participants sufficiently attended to the task (see methods). The participants played an online game in which they would explore and navigate a simulated museum consisting of multiple rooms. Because some of the rooms were clustered into local groups or "wings" of the museum, the subjects could learn a useful hierarchical

representation of the task. The participants were told to imagine that they were a tour guide in this museum who was required to guide visitors to specific paintings that they requested to see. In total there were 15 rooms in the museum, each containing a unique painting. Because the requested paintings occurred in different rooms, the participants needed to plan their route in order to reach the paintings in as few steps as possible. The 15 rooms were organized according to a "ring-of-cliques" layout as shown in Fig 1A, which was originally used by [30]. In this graph, each room is connected to four other rooms via four different hallways. This layout of the museum was never shown to the participants, nor did they receive any other hints about the layout. During an initial "training phase" (see Methods), the participants were required to learn the layout of the museum by engaging with the task. Then, during a subsequent "testing phase", which constitutes the main focus of our analyses, the participants were required to navigate the museum based on this learned knowledge.

The experimental task was divided into "miniblocks". At the beginning of each miniblock, the participant was cued with an image of a specific painting that they would be required to find in that miniblock. The participant could navigate from room to room in the museum by pressing either the <z> or the <m> key of their keyboard in order to select the direction that they wished to move next. However, each key mapped to two possible rooms with equal probability (Fig 1B). The mappings were engineered to provide the participant agency at the level of the wings of the museum: Each key consistently moved the participant out of the current wing in only one direction, allowing them to choose a direction to "rotate" through the wings of the museum. On reaching the goal painting for that miniblock, the participant was required to press the <space> key. They were told that if they did so correctly, then they would receive a small reward that would accumulate throughout the experiment as a bonus payment. However, if they failed to press <space> in the room with the goal painting, or if they pressed <space> in a room that was not the current goal, then they would instead incur a small financial punishment. These requirements ensured that the participants would actively remember the identity of the goal painting for that miniblock, in order to successfully find it and receive the reward. The miniblock ended once the room with the goal painting was reached, irrespective of whether the subject pressed <space> or not, and the next miniblock started with a new goal cue. The starting location for the subsequent miniblock was always the goal location of the last miniblock, so the entire experiment consisted of a single continuous walk through the museum. The new goal painting was always a room located in a wing different from the current starting location. Fig 1D provides an overview of the timeline of a single miniblock.

## Cognitive models

We fit multiple cognitive computational models (see below) to the response times and the choices, which we predicted would be sensitive to lower-order task structure and simplified hierarchical abstraction, respectively. We preregistered the exact structure of the models, including the definition of all theoretically meaningful and all "nuisance" regressors (https://osf.io/n2jcz/; with exceptions marked and justified in S2 Appendix). We simulated task performance for each of the proposed models to confirm that they make different predictions about behavior, and conducted a simulation study to validate the results for both our model comparison and parameter estimation techniques (see S1 Appendix). These simulations lend confidence that our main results reported here are meaningfully interpretable [39].

## Null model

In order to provide a benchmark for the other cognitive models, we included "null models" that capture behavioral predictions of no theoretical interest. For choice behavior, we

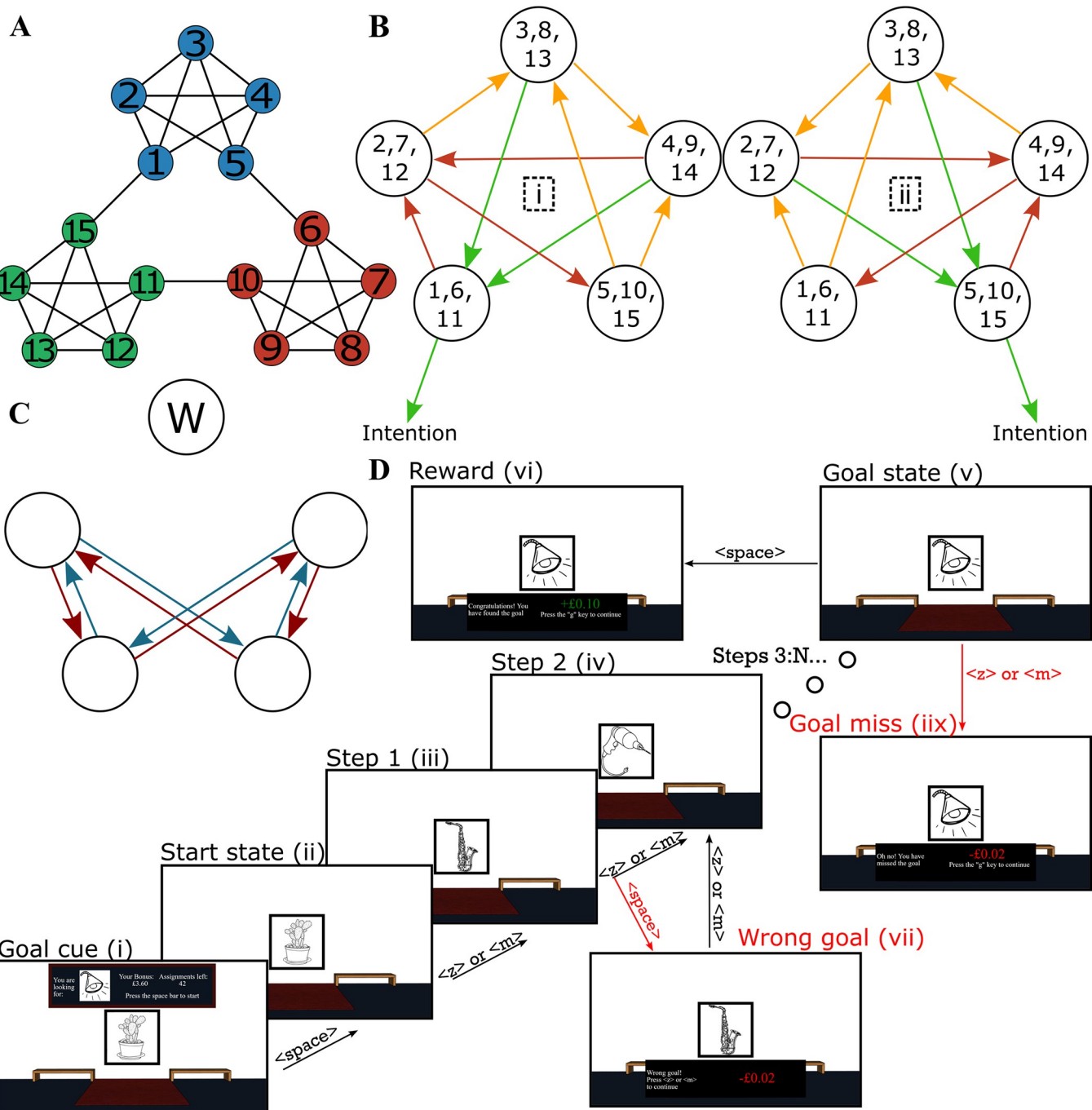

**Fig 1. Task design.** (**A**) A graph showing the relationships between the rooms in the museum task. The museum is characterized by three different communities or "wings", each consisting of 5 rooms. Individual rooms are numbered 1–15. (**B**) "Balanced" action-outcome mappings for each room. Each set of outcomes is randomly assigned to keys <z> and <m> (labelled "i" and "ii"). The same key mapped to the same set of outcomes across the three different wings. Hence, if the participant intended to move out of the current wing into a specific different wing, they needed only to select repeatedly the key leading to the desired wing (labeled as "intention"). Green arrows indicate transitions to the intended wing, orange arrows indicate transitions ending in nodes with green arrows, and red arrows indicate transitions ending in nodes without any green arrows. Note that the probabilistic mappings ensure that, even when the subjects select a response appropriate to their intended direction, the actual transition obtained might or might not be consistent with their intention (see S1 Appendix for more details). (**C**) Illustration of "preference" in action-outcome contingencies (see Methods). Blue arrows indicate transitions that are present for both actions (as shown in B). Red arrows indicate transitions that are not possible for either action. One room does not have any preferred or removed transitions. This room is labeled with "W" for "wide". (**D**) Illustration of an example miniblock. The first image is the goal cue (i). In this example, the participant begins the miniblock in a room with a cactus painting (center; images from ref (112) but illustrations shown here were sourced from openclipart.org), and the goal cue indicates that they must look for the "lamp" painting (top). The participant initiates the miniblock by pressing <space> (ii) (This "start state", which the

participants could anticipate based on the goal-cue (i) step, was excluded from the analyses). Subsequently, the participants were required to select either <z> or <m> to move between rooms (iii, iv) and <space> to indicate that they reached the goal (v), after which they received a reward (points later translated into a financial bonus) (vi). As illustrated, pressing <space> in a non-goal room results in a small penalty ("wrong goal") (vii). Similarly, pressing <z> or <m> in the goal room also leads to a small penalty ("goal miss") (iix). The miniblock always ends with either a "goal miss" (iix) or a "reward" (vi) screen, after which the next goal cue (i) is presented.

estimated response-coded logistic regressions where key <z> is coded as "0" and key <m> is coded as "1". The null model included only an intercept term and no regressors, accounting for a bias to select either <z> or <m> preferentially. For the response times, we estimated a linear model for the mean parameter of a log-normal distribution. The null model included six nuisance regressors and an intercept term. These nuisance regressors were also included in all of the other response time models. Firstly, we hypothesized that participants might get frustrated on longer miniblocks, causing them to speed up their response times. Hence, a nuisance regressor was included that captured the (log-transformed) number of "steps" for the current miniblock, indicating the number of button presses and transitions that were made that miniblock. Secondly, we hypothesized that participants might better remember rooms that were more recently encountered, causing them to speed up. Hence, a nuisance regressor was included that captured the (log-transformed) number of trials previously that the current room was last visited. Thirdly, we hypothesized that participants might be slower to execute responses on different keys compared to repetitions of the same key. Hence, a binary nuisance regressor was included that indicated whether the current key press was different from the previous key press.

The remaining three nuisance regressors were binary, each capturing an irregularity in the action-outcome contingencies. The reasoning behind these irregularities is further elaborated in the Methods section (section "action-outcome mapping") and in S1 Appendix. Here, we discuss how these irregularities might influence response times, and how the three nuisance regressors aim to capture this influence. Firstly, four transitions per community were possible regardless of which choice the participant made (Fig 1C, blue, "preferred"). Following a particular transition will elicit a prediction error, and larger prediction errors are hypothesized to yield slower responses for choices made in the subsequent state. Note that this is the basis for our prediction that participants should slow down following transitions between two different communities. Consider a participant who would behave completely randomly in the Museum task. In each room, they have a 50% chance of selecting either action, which yields a 50% chance of following either of two transitions as shown in Fig 1B. A preferred transition could be elicited by both actions with a 50% chance, and would thus be experienced more often. Therefore, it is likely these preferred transitions yield lower prediction errors (i.e. they are predicted more strongly), yielding faster responses for the choices immediately following the transition. To capture this, we added a binary regressor that was 1 following a preferred transition, and 0 otherwise.

Secondly, some transitions were not possible in our design (Fig 1C, red). Therefore, rooms with an outgoing red arrow as shown in Fig 1C have only three unique possible outcomes in total (each action has two outcomes, but one outcome is shared between the two actions as a preferred transition). Notice that there is one room that does not have any removed or preferred transitions (labeled with "W" for "wide" in Fig 1C). Because this is the only room that has 4 possible unique outcomes, participants might find it harder to decide on the most appropriate action in this room (as they have to consider more possible outcomes) predicting slower responses in this room. For this reason, we included a binary regressor that was 1 when participants occupied this wide room, and 0 otherwise.

Thirdly, transitions out of this wide room might be harder for participants to predict, given that each action has two unique outcomes as opposed to one unique outcome (with the other outcome shared between the two actions as a preferred transition in all other rooms). Outcomes that are harder to predict may yield larger prediction errors, and thus predict increased response times for the choices that immediately follow these transitions. For this reason, we included a binary regressor that was 1 following transitions out of this wide room, and 0 otherwise.

## Model-based

Among the cognitive models, we first considered a canonical model-based reinforcement learning agent. This agent was equipped with perfect knowledge of the action-outcome contingencies and could compute the exact (temporally discounted) values of each room with respect to the current goal. Parameter fitting included the discount factor, which is known to influence community structure [29]. In particular, moderately high values for the discount factor emphasize differences between the wings, increasing value inside the goal wing relative to values outside the goal wing. We hypothesized that participants might base their choices on the difference in expected values (EVs) between choices <m> over <z>, making the choice with the relatively higher value (corresponding to shorter paths to the goal) more likely to be selected. Hence, to model choice behavior, the model-based model included the relative difference of EV for <m> over <z> as a regressor. To model response times, the model-based model included three regressors besides the six nuisance regressors. Firstly, we hypothesized that participants might slow their responding when they are closer to the goal, in order to not accidentally trigger a "goal miss" (Fig 1D). Hence, the model-based response time model included a regressor for the expected value of the current step. Secondly, we hypothesized participants might slow their responses for unexpected progress toward or away from the goal. Hence, the model-based response time model included a regressor for reward prediction error (RPE), defined as the difference between the EV of the current step and the EV of the previous step. Thirdly, we hypothesized participants might slow their responses when the value difference between the two choices is less salient. Hence, the model-based response time model included a regressor capturing the (inverse) "conflict", which was evaluated as the absolute difference between the EV of the two available choices.

## Explicit hierarchical structure

Secondly, we considered a cognitive model that tested graph-theoretic predictions of hierarchical abstraction. This agent accessed a perfect representation of the mappings between the lower-level rooms and the higher-level wings, but not of the exact locations of the rooms within the wings (whereas the model-based agent represented the lower-level room locations, without any explicit indication of the mappings with respect to the wings). We hypothesized that the participants might repeatedly select the choice that directly "rotated" towards the current goal wing. Hence, the explicit hierarchical model of choice behavior included a binary regressor indicating whether or not <m> was the choice rotating toward the goal wing. This rotation corresponds to the hierarchically informed policy where only one key allows for a transition to one of the two other wings. To model response times, the explicit hierarchical model included three regressors besides the six nuisance regressors. Firstly, we hypothesized that participants might slow their responding when they are closer to the goal. Hence, the explicit hierarchical response time model included a binary regressor indicating whether or not the current room was in the same wing as the goal. This is analogous to the EV regressor in the model-based model. Secondly, we hypothesized that participants might slow their responding following a transition between two wings compared to transition within one wing.

This effect has been observed in previous experiments and has been defined as a signature of higher-order predictive learning [29,40,41]. Hence, the explicit hierarchical response time model included a binary regressor indicating whether or not the previous transition mapped between two wings. Thirdly, we hypothesized that the participants might slow their responding in rooms that allow for a transition between two wings, because here one choice will allow for the transition between wings whereas the other choice excludes this transition. Participants might deliberately seek out or avoid these transitions and thus take more time to consider an optimal choice. Hence, the explicit hierarchical response time model included a binary regressor indicating whether or not the current room allowed for a transition between two wings. This is analogous to the measure of (inverse) conflict defined in the model-based model.

## Successor representation

Finally, we considered an agent that learned a successor representation, which encodes for each state the expected future (discounted) states that the agent will occupy. The discount factor parameter determines the temporal horizon of the expected future state occupations. Moderately high discount factors emphasize the higher-order wing structure of the museum, whereas lower discount factors emphasize the lower-order room locations. Importantly, the successor representation differs from optimal model-based decision making in two key respects. Firstly, the successor representation holds a multi-step predictive model of future state occupations, while the model-based model holds a one-step predictive model. This explains why the successor representation can predict slowing between communities. It also implies generalization of value across different rooms in the same wing. Optimal model-based decision making requires more intensive computation to deal with changing goals [18,25] and would predict perfect adherence to the values of individual rooms in each wing. Secondly, the successor representations' predictive model of states is dependent on the policy being followed. Because different goal locations imply different optimal choices in our museum task, the internal model of the task states will be in constant flux across successive task miniblocks, even after a training phase in which the participants learned a good baseline model of the environment [42]. This differs from the model-based model, which is provided a priori with an accurate model of the environment that can be used to compute an optimal policy for any possible goal location. After each transition, the successor representation yields a "state prediction error" that corresponds to whether the newly observed state was anticipated or not. This prediction error is used to update the predictive model according to a temporal-difference learning rule (see Methods). The predictive representation can be used to compute expected values analogous to those computed by the model-based model. To model choice behavior, the successor representation model included a similar regressor as for the model-based model, reflecting the relative difference in expected values between the two choices. To model response times, the successor representation model included similar regressors reflecting the EV, the RPE, and the conflict. We additionally hypothesized participants might slow their responding when encountering more surprising rooms. Hence, in contrast to the model-based model, the successor representation model included a regressor for state prediction error. Dependent on the discount factor, state predictions are expected to be larger for transitions between two wings compared to transitions within one wing. Therefore, this state prediction error regressor is analogous to the binary regressor for transitions between wings in the explicit hierarchical model.

## Choice behavior

All analyses with respect to choice behavior and response switching were preregistered (https://osf.io/n2jcz/) and thus of confirmatory nature. Random-effects group-level Bayesian

model selection of the computational models of choice behavior indicate a significant group difference in model prevalence ($BOR < 10^{-26}$) with a strong preference for the explicit graph-theoretic hierarchical abstraction model ($pxp = 1$). This holds when considering binary model comparisons between the explicit model and every other model (all $BOR < 10^{-5}$, see Methods for multiple comparison correction. All $pxp = 1$). Posterior model probabilities for every participant are shown in Fig 2A.

We tested the group-level significance of the relevant rotation-choice parameter of the explicit hierarchical model by fitting a hierarchical Bayesian logistic regression with separate participant-level effects (Fig 2B). We find that participants are indeed significantly more likely to choose the action that leads into the rotational direction of the goal wing ($M = 1.172$, $HDI_{95\%} = [0.992, 1.350]$, $ER_+ = \infty$). Fig 2C shows the difference in proportion of <m> choices when <m> is the correct rotational direction versus when <z> is the correct rotational direction. A clear effect can be seen in the data (blue) that is also well captured by the explicit graph-theoretical abstraction model (red). It can also be seen that the effect of rotational direction appears smaller for <z> than for <m>. Indeed, the intercept in the hierarchical Bayesian logistic regression is significantly above 0 ($M = 0.311$, $HDI_{95\%} = [0.220, 0.401]$, $ER_+ = \infty$) indicating a bias in selecting <m>. This can be explained by the fact that <m> is placed on the right side of the QWERTY-keyboard and participants might have a bias to interpret rightward directions as progressive (i.e., reading in most western cultures happens from left to right), or the fact that the majority of participants are likely to be right-handed (which we did not assess).

## Response switches

Given that participants preferentially selected the key corresponding to the rotational direction of the goal wing, we examined whether participants also dynamically updated their responses during task execution. We tested this by asking whether they were more likely to switch their response immediately after moving out of the goal wing, indicating an intention to reverse their direction of movement (i.e., go "backwards"). Similarly, we asked whether participants were less likely to switch their response immediately after moving into the goal wing, indicating an intention to continue in the same direction. We modeled this with a hierarchical Bayesian logistic regression, the coefficients of which are shown in Fig 2D. The model confirms significant effects for goal wing entry indicating decreased switching ($M = -0.722$, $HDI_{95\%} = [-0.940, -0.509]$, $ER_- = \infty$), goal wing leave indicating increased switching ($M = 0.983$, $HDI_{95\%} = [0.581, 1.349]$, $ER_+ = \infty$), and a nuisance regressor for action repetitions ($M = -0.724$, $HDI_{95\%} = [-0.916, -0.532]$, $ER_- = \infty$), which indicated that participant switched less often when they had already repeated an action multiple times (i.e., action commitment). Fig 2E illustrates the proportion of response switches relative to "baseline" for all trials with an entry into the goal wing ("entry"), vs all trials with an exit from the goal wing ("leave"). The baseline trials include all trials that are not classified as entry or leave. The empirical results (blue) indicated that participants were less likely to switch when they entered the goal wing, and were more likely to switch immediately after leaving the goal wing. Posterior predictions from the hierarchical Bayesian logistic regression model match these data well (red).

## Participants mix abstraction with detail

For each wing, given a goal in each other wing, there was always one room where the optimal choice according to a model-based (and converged successor representation) model would deviate from the explicit hierarchical model. This is because both the rotation and the antirotation can lead to the same desired outcome, but the rotation has a chance of transitioning to a

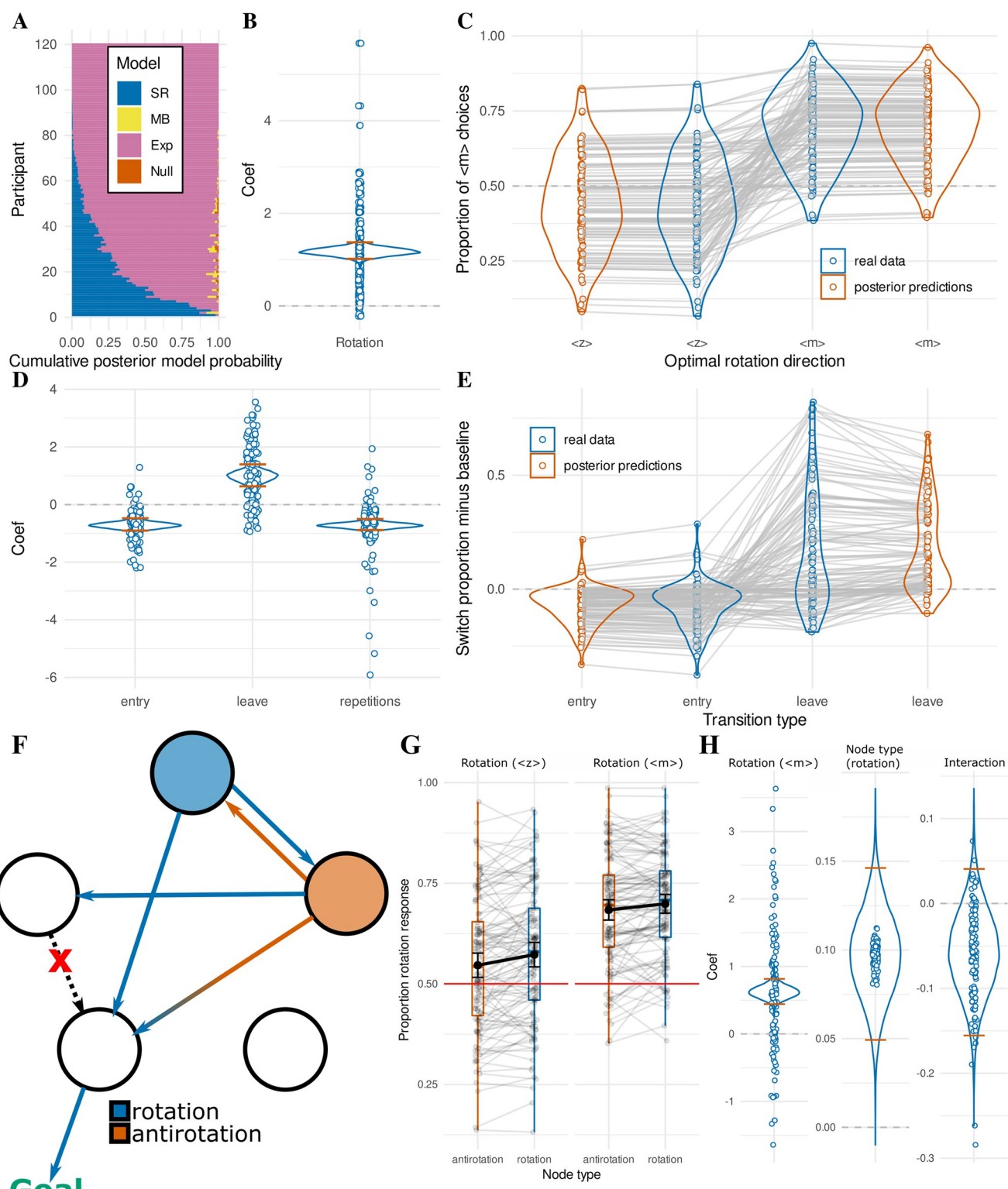

**Fig 2. Choice data.** (**A**) Horizontal stacked bar charts for every participant, illustrating the posterior model probability derived from random effects Bayesian model comparison between four cognitive models, "Null" (orange), explicit hierarchical "Exp" (pink), model-based "MB" (yellow), and successor representation "SR" (blue). Participants were sorted by posterior model probability for the "Exp" model for interpretability. (**B**) Regression coefficient of the "rotation" variable in the explicit hierarchical logistic regression. Density plot shows the full posterior distribution of the population mean, with orange lines indicating the 95% highest density interval. Dots represent posterior means of individual participant (random) effects. (**C**) Proportion of <m>

choices over the course of the testing phase, grouped for every participant by trials where <m> was the optimal rotation vs where <z> was the optimal rotation. Empirical data are shown in blue. Posterior predictions from the explicit hierarchical logistic regression are shown in orange. (**D**) Regression coefficients for the response switch analysis, describing the probability of switching response key upon "entry" of the goal wing, "leave" of the goal wing, and the number of "repetitions" of pressing the current key. Colors similar to B. (**E**) Proportion of trials on which the participant switched their response key, grouped by trials where the participant just entered the goal wing or just left the goal wing. Switch proportion of all other ("baseline") trials is subtracted, so values below 0 reflect a decreased tendency to switch, and values above 0 an increased tendency. Color coding as in C. (**F**) Given a particular goal outside the current wing, one action corresponds to the correct "rotation" to follow for most of the rooms in the current wing. However, one room in each community actually leads to better outcomes when the participants would follow the opposite action (antirotation, here colored in orange). In this figure, possible outcomes of the rotational action are colored blue, and possible outcomes of the antirotational action are colored orange. As can be observed, the orange room has a preferred transition in the direction of the goal wing (colored with an orange to blue gradient, indicating both choices can lead to this outcome). Interestingly, when following the rotation action (blue transitions), one possible transition leads in the correct direction whereas the other possible transition leads to a room for which the transition in the direction of the goal wing was removed (X in the figure). By contrast, when following the antirotation (orange transitions), the other possible outcome leads to a state that still has a transition in the direction of the goal wing (blue circle). For this reason, model-based and (converged) successor representation models predict participants would pick the antirotational action in this orange room, and the rotational action in the blue room. (**G**) Proportion of rotational actions chosen in the orange (antirotation) and blue (rotation) rooms ("Node type", x-axis, colored as in F), considered separately when the correct rotation would be <z> (left panel) or <m> (right panel). Black dots with error bars correspond to mean with 95% confidence interval. (**H**) Regression coefficients for the (anti)rotation selection analysis, describing the probability of selecting the correct rotation when it is <m> (over <z>, left) and when occupying the blue room (as opposed to the orange room, see F; middle). The interaction term is displayed on the right. Colors similar to B.

room without connections towards the goal wing, whereas the antirotation does not (Fig 2F). We set up an exploratory hierarchical Bayesian logistic regression to test whether participants are indeed more likely to follow the rotation when it is the optimal decision (blue, Fig 2F) compared to when it is not (orange, Fig 2F). Fig 2G shows that in both of these rooms, regardless of which direction the rotation follows (<z> or <m>), participants select the correct rotation on most of the trials, but they do so less often when the antirotation offers a better alternative. A test of the intercept of the logistic regression confirms that participants are indeed overall biased towards selecting the rotation, as predicted by the explicit hierarchical model ($M = 0.580$, $HDI_{95\%} = [0.489, 0.671]$, $ER_+ = \infty$). In accordance with our earlier observations (Fig 2C), participants are more likely to do so when the correct rotation is <m>, since they have a bias towards selecting this response ($M = 0.621$, $HDI_{95\%} = [0.435, 0.809]$, $ER_+ = \infty$; Fig 2H, "rotation"). Crucially, when participants occupy the room where the rotation is the optimal direction (blue, Fig 2F), they are more likely to select the rotation, compared to when the antirotation is in fact optimal (orange, Fig 2F) ($M = 0.097$, $HDI_{95\%} = [0.048, 0.145]$, $ER_+ = 9999.000$; Fig 2H, "node type"). This effect is numerically, although not significantly, attenuated when the correct rotation is <m> ($M = -0.057$, $HDI_{95\%} = [-0.156, 0.040]$, $ER_- = 7.061$; Fig 2H, "interaction"). Overall, this confirms that participants' choice behavior is mostly explained by an explicit hierarchical model in agreement with our model comparison, as evidenced by the large influence of the rotation even when this action is not in fact optimal. However, small influences of more precise (model-based or successor-like) knowledge is evident at the group level, indicating that the best model might allow for a mixture of hierarchical abstraction and fine-grained knowledge [43].

## Response times

All analyses with respect to response times described here were preregistered and thus of confirmatory nature (https://osf.io/n2jcz/). Random-effects group-level Bayesian model selection of the computational models of the response time data indicate a significant group difference in model prevalence ($BOR < 10^{-10}$) with a strong preference for the successor representation model ($pxp = 1$). This holds when considering binary model comparisons between the successor representation and every other model (all $BOR < 0.003$, see Methods for multiple comparison correction. All $pxp > 0.998$). Posterior model probabilities for every participant are shown in Fig 3A.

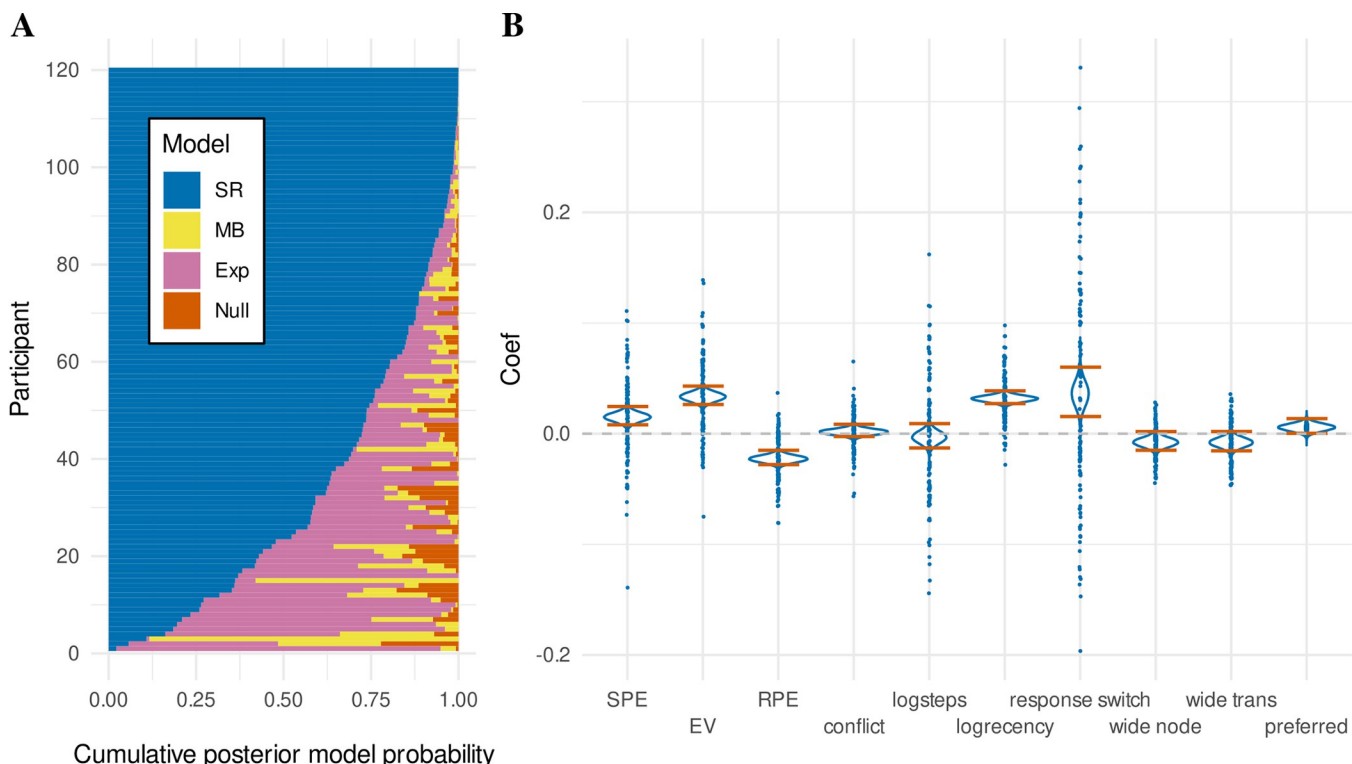

**Fig 3. Response time data.** (**A**) Horizontal stacked bar charts for every participant, illustrating the posterior model probability derived from random effects Bayesian model comparison between four cognitive models, "Null" (orange), explicit hierarchical "Exp" (pink), model-based "MB" (yellow), and successor representation "SR" (blue). Participant data were sorted by posterior model probability for the "SR" model for interpretability. (**B**) Regression coefficients of the successor representation model. Density plots show the full posterior distributions of the population means, with orange lines indicating the 95% HDI. Dots represent posterior means of individual participant (random) effects. SPE refers to the state prediction error, EV to expected value, and RPE to reward prediction error. See text for definition of remaining terms.

We tested whether the estimates of the nuisance regressors included in the successor representation model were indeed significantly related to response times (Fig 3B). We do not find any evidence that participants speed up or slow down based on the (log-transformed) number of trials on the current miniblock ("logsteps") ($M = -0.003$, $HDI_{95\%} = [-0.014, 0.007]$, $ER_- = 2.510$). We do find a significant effect of response slowing for rooms based on (log-transformed) recency ("logrecency"), indicating rooms that have not been encountered for longer yield slower responses ($M = 0.032$, $HDI_{95\%} = [0.026, 0.037]$, $ER_+ = \infty$). We also find significant slowing for trials where the participant switched their response key ("response switch") ($M = 0.036$, $HDI_{95\%} = [0.014, 0.059]$, $ER_+ = 1378.310$).

Further, we find trending effects for an influence of the action-outcome mapping (see Methods) on response times (Fig 3B). In particular, we find a trend for participants to be faster in rooms that have more possible outcomes ("wide node", which have no "preferred" transitions) ($M = -0.008$, $HDI_{95\%} = [-0.016, 0.001]$, $ER_- = 28.390$) and the transitions following these rooms ("wide trans") ($M = -0.008$, $HDI_{95\%} = [-0.017, 0.001]$, $ER_- = 26.615$). Meanwhile, we find a trend for participants to be slower following more expected transitions, which occur when both actions can transition to the same room (Fig 1C) ("preferred") ($M = 0.006$, $HDI_{95\%} = [-0.001, 0.012]$, $ER_+ = 20.158$). Parenthetically, these trends run opposite to our initial expectations, although they are not statistically reliable according to our definition, and reverse direction in a control model considering a state-based successor representation (see S2 Appendix).

With respect to the regressors of theoretical interest (Fig 3B), we find significant slowing for transitions yielding higher state prediction error ("SPE"), indicating slowing after transitions between wings compared to transitions within wings ($M = 0.015$, $HDI_{95\%} = [0.007, 0.023]$, $ER_+ = 6152.846$). We also find significant slowing for higher expected values ("EV"), indicating that participants respond slower for rooms closer to the current goal ($M = 0.033$, $HDI_{95\%} = [0.025, 0.042]$, $ER_+ = \infty$). We also find significant slowing for transitions with more negative reward prediction errors ("RPE"), consistent with post-error slowing when participants move away from the goal, or conversely, speeding when participants move toward the goal (yielding positive reward prediction errors) ($M = -0.023$, $HDI_{95\%} = [-0.029, -0.016]$, $ER_- = \infty$). Finally, we find no conclusive evidence that the regressor for conflict, conceptualized as the difference in expected value between the two different actions, influences response times ("conflict") ($M = 0.002$, $HDI_{95\%} = [-0.004, 0.007]$, $ER_+ = 2.514$).

## Posterior predictive checks

We further explored the successor representation model by examining specific transition types (Fig 4A). As illustrated in Fig 4B, which shows the regressor values assigned by the successor representation model to these trial types (extracted from the maximum a-posteriori fit for each participant), the three transition types that cross between wings ("between", "outof", and "into") yielded increased state prediction errors ("SPE"), as expected. Further, transitions that end in the goal wing ("inside" and "into") yield larger expected values ("EV"), transitions out of the goal wing ("outof") yield larger negative reward prediction errors ("RPE"), and transitions into the goal wing ("into") yield larger positive reward prediction errors. As a validation of our task design, we also replicated previous accounts of response time slowing for between-wing transitions [29,40,41] by conducting a paired t-test of "outside" (green) and "between" (blue) transitions, which approximately controls for effects related to value (EV, RPE, and conflict). We find significant slowing for between-wing transitions ($M_{between>outside} = 34.797$ms, $SD_{between>outside} = 85.459$, $t(118) = 4.442$, $p < 10^{-4}$), comparable to these previous reports.

Fig 4C plots the mean response times for participants separately for each trial type (left), and matching mean response times for the full posterior predictive distribution (right). It can be seen the model predicts response times in a very similar range as the real data, and mostly preserves the order of different trial types. However, there appears to be a trade-off whereby transitions "inside" the goal wing appear slightly overestimated, and transitions "into" the goal wing appear slightly underestimated. Additional exploratory modeling reported in S2 Appendix suggests this can be accounted for by modeling the absolute (unsigned) reward prediction error, as opposed to the signed reward prediction error.

Even though the successor representation model accounts well for the response time effects observed between the different transition types illustrated in Fig 4A, Fig 4B indicates substantial variance in the regressor values between trials of the same transition type, including overlap between the trials of different transition types. To investigate whether this variance in regressor values is also reflected in response times, we ordered all trials by regressor value (separately for each regressor) and binned them by intervals of 0.3. This yields a response time distribution for each bin; Fig 4D shows the mean (orange) and the 10th and 90th percentile (blue) for each bin. This reveals a parametric relationship between response times and the regressors derived from the successor representation, where for example the response times gradually increase in association with increasing expected value. This fine-grained analysis provides additional support that participants learn a detailed, predictive representation of the room locations, beyond just a simplified (explicit hierarchical) representation of the relationships between the three different wings.

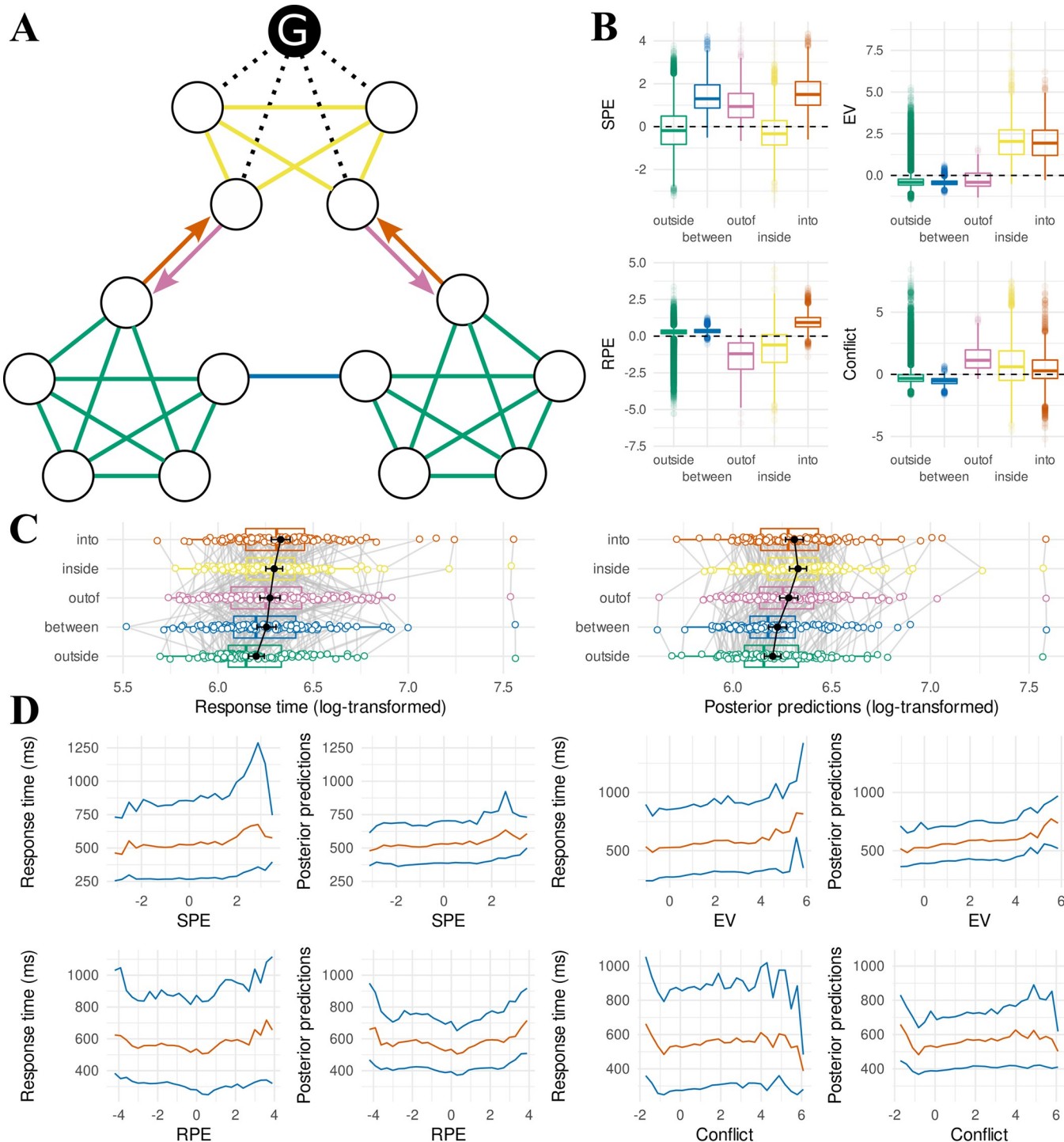

**Fig 4. Posterior predictive checks of the response time successor representation model between communities.** (**A**) Layout of the museum with transitions colored according to type. The goal painting is colored black and labeled "G". Green transitions stay within wings that do not contain the goal painting ("outside"). Blue transitions move between non-goal wings ("between"). Orange transitions move from a non-goal wing into the goal wing ("into"). Pink transitions move from a goal wing into a non-goal wing ("out of"). Yellow transitions stay within the goal wing ("inside"). (**B**) Distributions of regressor values assigned to trials following the different transition types as defined in A. Regressor values were computed based on the discount factor value at the maximum a-posteriori of the joint likelihood of the participant-level successor representation model fit. SPE: state prediction error, EV: expected value, RPE: reward prediction error. (**C**) Mean (log-transformed) response times (left) and mean posterior predictions (right) for each participant for the different transition types as defined in A. Population means with their 95% confidence interval are shown in black. (**D**) Mean (orange) and 10th and 90th percentile (blue) of the response time distribution for trials binned by regressor values (as assigned in B) in intervals of 0.3. Binned separately for the different regressors (SPE, EV, RPE, conflict). Conditional response time distributions are repeatedly plotted side by side for real data (left) and the full posterior predictive distribution (right).

To probe further how the successor representation might explain response times at a fine-grained (i.e. within-wing) level, we partitioned the trials as shown in Fig 5A, indicating whether the current room lies at a boundary allowing for a transition to a different community and specifying whether this transition would correspond to entering the goal community (orange), leaving the goal community (yellow), or switching between non-goal communities (blue). These trials can be compared against trials within the goal community (pink) or outside of it (green). To ensure the variance in response times on these trials corresponds to knowledge of the within-community structure of the museum, we excluded all trials that followed a between-community transition. We conducted three paired t-tests to investigate sensitivity to the within-community structure of the museum (corrected for multiple comparisons; see Methods). With respect to rooms that are outside the goal community (Fig 5B), we found a significant increase in response times in rooms at the boundary that allows for a transition between two non-goal communities (blue), compared to non-boundary rooms outside the goal community (green) ($M_{non\text{-}goal>outside}$ = 6.921ms, $SD_{non\text{-}goal>outside}$ = 32.292, $t(119)$ = 2.348, $p$ = 0.041). We also find a significant increase in response times for rooms at the boundary that allows for a transition "toward" the goal community (orange) compared to rooms at the boundary that allow for a transition between non-goal communities (blue) ($M_{toward>non\text{-}goal}$ = 10.714ms, $SD_{toward>non\text{-}goal}$ = 42.671, $t(119)$ = 2.751, $p$ = 0.021). Fig 5C shows posterior predictions from the successor representation model. This reveals the model predicts a similar increase in response times for boundary-rooms that allow for transitions into the goal community (orange), but not for boundary-rooms that allow for transitions away from the goal community (blue; compared to non-boundary rooms outside the goal community, green). Inspection of the boundary and non-boundary rooms in the goal community (Fig 5D) did not reveal statistically significant differences in response times ($M_{outof>within}$ = 6.445ms, $SD_{outof>within}$ = 79.123, $t(119)$ = 0.892, $p$ = 0.374), nor does the model predict any such differences (Fig 5E).

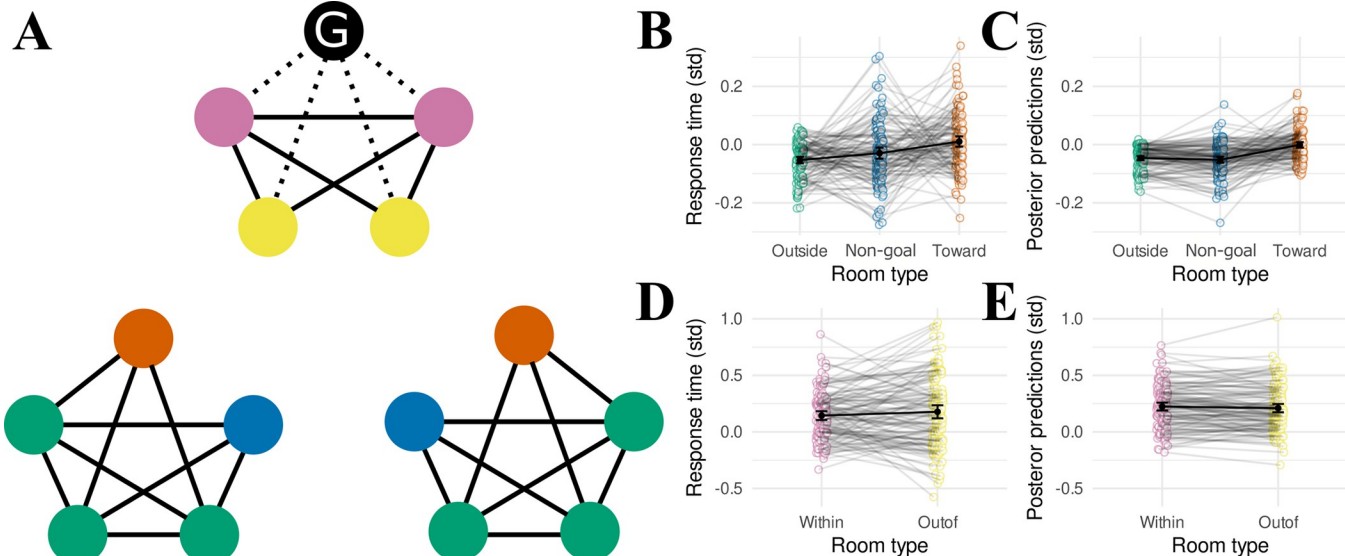

**Fig 5. Posterior predictive check of the response time successor representation model within communities.** (**A**) Layout of the museum with rooms colored according to type. The goal painting is colored black and labeled "G". Green rooms lie within wings that do not contain the goal painting ("outside"). Blue rooms lie at the boundary between two non-goal wings ("away"). Orange rooms lie at the boundary allowing for a transition into the goal wing ("toward"). Pink rooms lie within the goal wing ("within"). Yellow rooms lie at the boundary allowing for a transition out of the goal wing ("out of") (**B**) Mean response times (standardized within participant) in different rooms outside the goal wing, as labeled in A. (**C**) Same as B, but drawn from posterior predictions of the fitted successor representation for each participant. (**D**) Same as B, but for rooms inside the goal wing. (**E**) Same as D, but drawn from posterior predictions of the fitted successor representation for each participant.

## Control analyses and convergence

We conducted several control analyses to check the robustness of the above results, and replicated all results in all cases (see S2 Appendix). Additionally, we explored an alternative formulation of the successor representation that only tracked state-state predictions [23], rather than state-action conjunctions [24,26]. We found that these two models both perform significantly better than the other models (null, explicit-hierarchical, and model-based), but do not significantly differ from each other (see S2 Appendix). Exploring the state-action successor representation further, S1 Appendix shows that the successor representation is gradually learned over the course of the training phase, and has saturated by the time the testing phase starts. Additionally, S1 Appendix shows that the variance explained by the local surprise (as indexed by the "log-recency" nuisance regressor) is independent from the variance explained by the global surprise (as indexed by the successor representation prediction error).

## Free sort post test

We investigated whether the participants learned an accessible representation of the structure of the museum by asking the participants after the testing phase to reconstruct the layout of the museum by arranging the paintings on a grid. The Euclidean distances between the positions of all possible painting pairs (105 per participant) were then computed and standardized within participants. Fig 6A shows the individual distances and average distances for all painting pairs, classified according to four different categories. A hierarchical Bayesian linear regression (preregistered; https://osf.io/n2jcz/) indicated that participants on average placed paintings that were part of the same community closer together ($M$ = -0.578, $HDI_{95\%}$ = [-0.699, -0.460], $ER_-$ = $\infty$) (Fig 6B), as reflected in low distances attributed to "community-

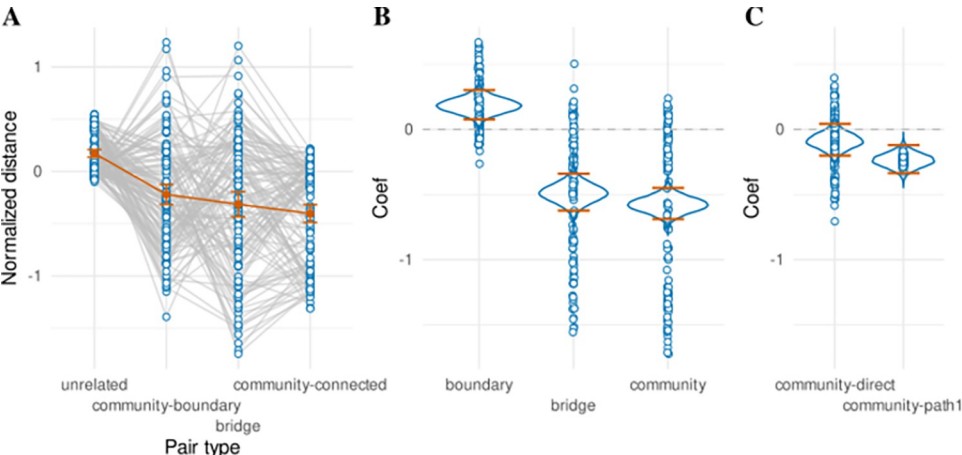

**Fig 6. Free sort post-test results.** (**A**) Euclidean distances (normalized) for different pairs of paintings, grouped by whether they were part of the same community and directly connected (community-connected) or not (community-boundary), or whether they were directly connected but not part of the same community (bridge), or not connected and not part of the same community (unrelated). Orange dots with error bars represent mean with 95% standard error. (**B**) Regression coefficients of the free sort regression. Density plot shows the full posterior distribution of the population mean, with orange lines indicating the 95% highest density interval. Dots represent individual participant posterior means. Note that the "community" effect captures both "community-connected" and "community-boundary" effects as shown in A, and that the "boundary" effect captures their difference. (**C**) Effects that quantify bias induced by community structure. The "community-direct" effect only investigated directly connected rooms, and asked whether paintings of the same community (community-connected in A) were placed closer together than those of different communities (bridge in A). The "community-indirect" effect is analogous, but investigating pairs of rooms that had a minimal distance of one intermediate room. This corresponds to "community-boundary" paintings in A, and a subset of "unrelated" paintings.

connected" paintings (Fig 6A). However, this effect was attenuated for paintings that were not directly connected (i.e. between the two paintings that lie at the "boundary" of each community, Fig 6B) ($M = 0.183$, $HDI_{95\%} = [0.071, 0.295]$, $ER_+ = 1051.632$), consistent with higher distances reported for these relationships ("community-boundary", Fig 6A). The participants also placed paintings that were directly connected but not part of the same community (i.e. between-community transitions or "bridges", Fig 6B) closer together ($M = -0.489$, $HDI_{95\%} = [-0.633, -0.349]$, $ER_- = \infty$), as shown by the lower distances for "bridge" relationships (Fig 6A). This pattern of results is suggestive of fully accurate reconstruction of the graph, since coefficients associated with all lower-order relationships indicate significant effects. To test for specific bias introduced by the community structure (as preregistered), we checked whether participants placed paintings that are connected and part of the same community closer together than paintings that are connected but not part of the same community ("bridges"). Although this analysis yielded an effect indicative of bias, the result was not statistically reliable ($M = -0.089$, $HDI_{95\%} = [-0.191, 0.013]$, $ER_- = 12.387$; Fig 6C, "community direct").

We ran an additional exploratory test to isolate the effect of community structure from the effect of direct connections [44]. A hierarchical Bayesian linear regression that included regressors only for pairs of paintings with the shortest possible distance of 1 intermediate room between them indicated that the participants placed paintings with a path distance of 1 closer together if they were part of the same community (i.e. the two paintings at the boundaries of one community) compared to when they were part of different communities ($M = -0.238$, $HDI_{95\%} = [-0.347, -0.131]$, $ER_- = \infty$) (Fig 6C, "community-indirect").

## Successor representation modularity is associated with better hierarchical structure learning

The degree that a successor representation accounts for community structure depends on the discount factor parameter for that model. We quantified this by computing the ratio of the state prediction error yielded by a transition between two wings, over the state prediction error yielded by a transition within a wing (controlling for "preferred" transitions; see Methods). We called this measure the "modularity". A modularity of 1 indicates similar state prediction errors for transitions between wings and within wings, whereas modularities greater than 1 indicate larger state prediction errors for transitions between wings. For example, a modularity of 2 would indicate the state prediction error for a transition between wings is twice as large as for a transition within one wing. This modularity measure depends on the value of the discount factor, but also on the experienced sequence of states. Fig 7A and 7B) plot the modularity across a range of possible discount factors, after a sequence of states as experienced by two representative participants in our dataset.

Fig 7C shows the recovered posterior mean discount factors for the response time successor representation model. Although the distribution peaks around a value of 0.875, the density is relatively dispersed, decreasing as it approaches 0. Peaks close to (but not exactly at 1) correspond to successor representations that most strongly capture the community structure of the museum environment (e.g. Fig 7A). We computed the expectation of our modularity measure and tested whether it was significantly larger for the posterior distributions obtained for each participant as compared to a null model (a uniform prior over discount factors) (Fig 7D). A paired t-test confirmed a significant increase in modularity ($M_{fit>null} = 0.132$, $SD_{fit>null} = 0.182$, $t(119) = 7.977$, $p < 10^{-11}$). Note that the null model also predicts some degree of modularity, since it reflects a uniform distribution over the discount factor, including values that reflect higher modularity. An example of a modular successor representation as estimated for one of our participants is shown in Fig 7E.

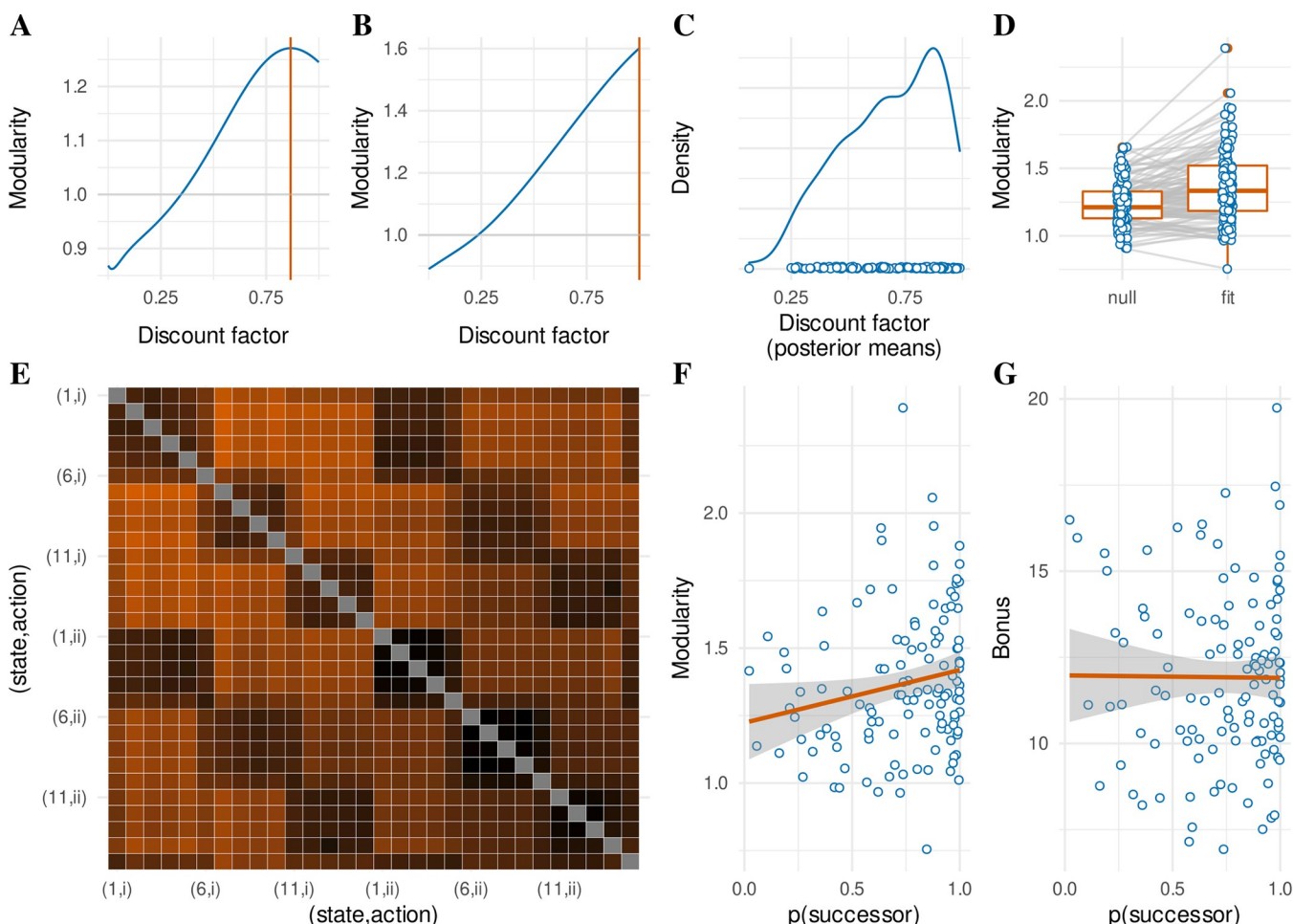

**Fig 7. Modularity of the successor representation model of response times.** (**A,B**) Relationship between possible discount factor values and "modularity" of the successor matrix after a sequence of states as experienced by two representative participants. Modularity is measured as the ratio of state prediction error for between-wing transitions over within-wing transitions (corrected for "preferred" transitions; see Methods). Notice that A shows a "peak" (orange line) just before 1, whereas B linearly increases with a maximum at 1. These two patterns were ubiquitous in our data set. Not all discount factors lead to modular successor representations (horizontal grey line at modularity = 1). For both these participants, only discount factors above about 0.25 show an effect of community structure. (**C**) Posterior means of the recovered discount factor parameters for all participants (dots) and kernel density estimate over these. (**D**) Modularity of the successor matrix for all participants (blue dots) under a null model and under the fitted successor representation model. (**E**) State prediction errors for all state-action transitions (even those not possible in the actual experiment), based on the estimated successor matrix for an example participant in our dataset (derived from the full posterior distribution; mean estimated discount factor of 0.978). Hotter colors indicate increased prediction error. Both axes index state-action conjunctions, with states labeled 1–15 as in Fig 1A, and actions labeled as (i) or (ii) as in Fig 1B. The community structure is visible as 'squares' of decreased prediction error for states (rooms) that are part of the same community (wing). (**F**) Posterior model probabilities of the successor representation response time model (x-axis) and the modularity measure (y-axis) for all participants (blue dots) with line of best fit (orange). (**G**) Similar to F with total accumulated reward bonus on the y-axis.

A preregistered (https://osf.io/n2jcz/) analysis indicated that across participants, higher estimated modularities were predicted by a higher posterior model probability for the successor representation model of response times (simple linear regression: β = 0.196, $F(1,118) = 4.468$, $R^2 = 0.036$, $p = 0.037$; Fig 7F). Note that this is a test between two measures both derived from the same fit of the successor representation model of response times. It thus provides insight mainly into the patterns of participant behavior that are uniquely explained by the successor representation model over the competing models (null, model-based, and explicit hierarchical), but not into whether successor representation learning can be observed in independent measures of task performance. By contrast, an additional preregistered regression

analysis asked whether the successor representation fit significantly predicted the performance of participants as expressed in terms of their total accumulated bonus payments. However, this analysis did not show evidence for any such relationship ($\beta$ = -0.073, $F(1,118)$ = 0.007, $R^2 <$ 0.001, $p$ = 0.935; Fig 7G).

Although there was strong evidence in favor of a successor representation model of response times across most individuals (see Fig 3A), there were also large individual differences in the modularity of these successor representations. Note that the modularity is directly derived from the state prediction error metric. Hence, the degree of modularity of the successor representation for a specific participant could be completely driven by the relative slowing on between-community transitions (compared to within-community transitions). Additionally, the estimated modularity of the successor representation could be informed by response time variance within communities, such as the slowing for rooms at the boundaries of a community as reported in Fig 5. This implies that modularity is in fact a more informative measure than response time slowing. We conducted a series of exploratory analyses that predicted independent task performance metrics based on the modularity of the response time-fitted successor representation models. To control for pure between-community response time slowing, we constructed hierarchical regression models where we first entered between-community response time slowing as the sole independent variable, and compare this against a second regression model including both response time slowing and modularity as independent variables. Between-community response time slowing was defined as the difference in mean response times between "outside" (green) and "between" (blue) transitions as indicated in Fig 4A. The modularity of the successor representation is defined as the ratio of the state prediction error elicited by between-wing transitions over within-wing transitions (corrected for "preferred" transitions; see Methods). This tests whether the modularity of the successor representation provides information above and beyond the between-community response time slowing. In total, this exploratory analysis yielded 15 regression models (all individual panels of Fig 8). When reporting the second-step model, we report the $F$ and $R^2$ statistics of the model itself, the slope of the newly added modularity regressor, and the $\Delta F$ statistic corresponding to the model comparison between the first- and second-step regression models. The reported p-value corresponds to this $\Delta F$ statistic. We corrected p-values for multiple comparisons across all first-step regression models and all comparisons with the second-step regression models (15 tests in total; see Methods for multiple comparison correction).

Our exploratory analysis confirmed a significant positive relationship between response time slowing and modularity ($\beta$ = 0.001, $F(1,117)$ = 7.472, $R^2$ = 0.060, $p$ = 0.044; Fig 8A). Contrary to the posterior model probability of the successor representation (Fig 7G), between-community response time slowing significantly predicted total accumulated reward bonus ($\beta$ = 0.005, $F(1,118)$ = 6.485, $R^2$ = 0.052, $p$ = 0.049; Fig 8B, top). Including the modularity of the successor representation as a second independent variable leads to a significant improvement in model fit ($\beta$ = 3.433, $F(2,117)$ = 12.435, $R^2$ = 0.175, $\Delta F(1,117)$ = 17.478, $p < 0.001$; Fig 8B, bottom). Note that the total bonus payment depended solely on the participant reaching a new goal in the minimum number of steps possible. The time taken to generate these choices (the response time) was not related to reward, as participants could perform the task entirely at their own pace. Instead, this relationship is suggestive of a link between successor representation learning and task performance, where participants who are more sensitive to the community structure of the museum generate more efficient paths to the goal.

In accordance with the positive relationship between response time slowing and reward, we observed a significant positive relationship between response time slowing and the posterior means for the explicit hierarchical "rotation" regressor of the choice data ($\beta$ = 0.003, $F(1,117)$ = 14.281, $R^2$ = 0.109, $p$ = 0.002; Fig 8C, top), indicating that participants whose response times

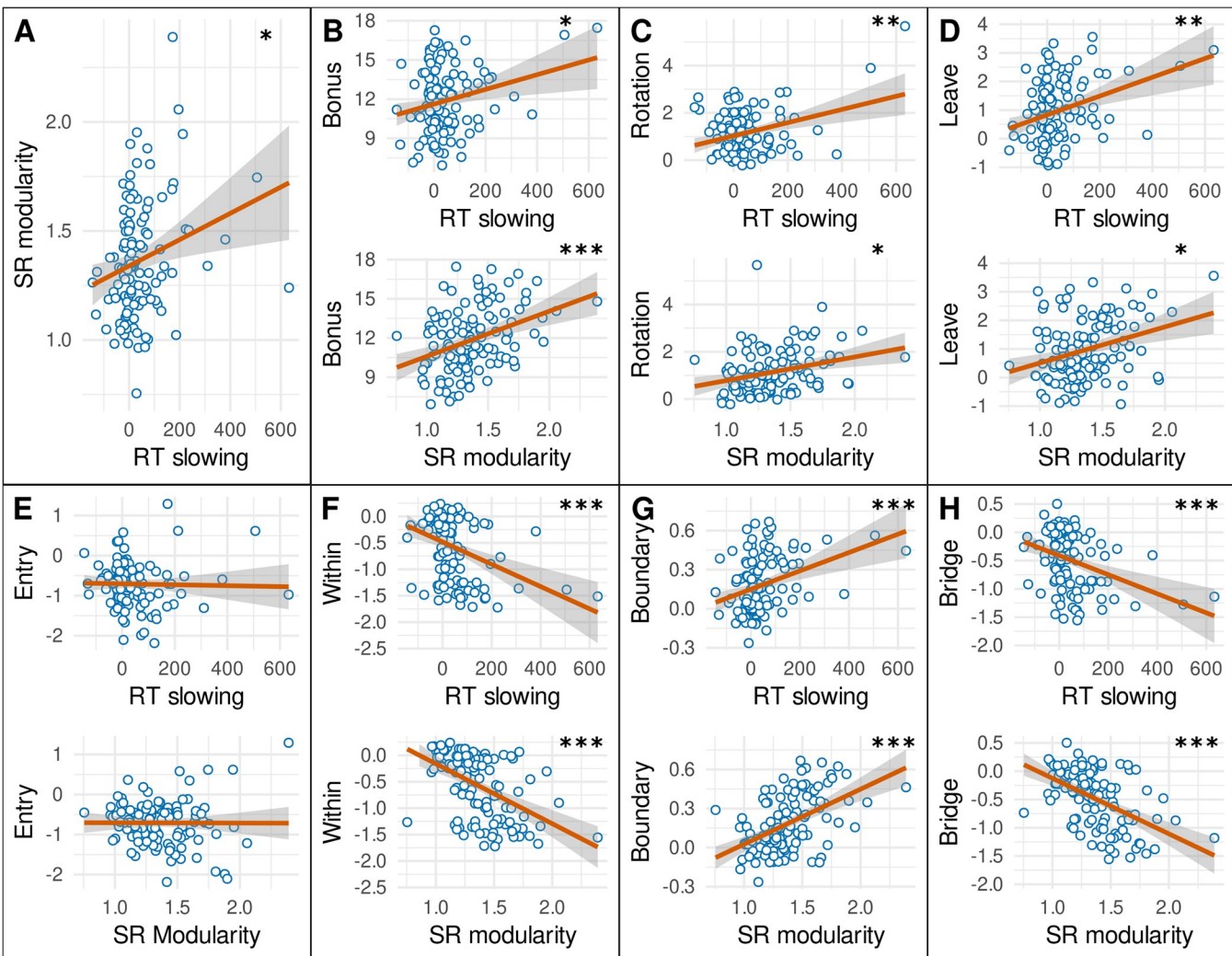

**Fig 8. Modularity correlations with task performance.** (**A**) Response time slowing (x-axis) for transitions between communities (blue in Fig 4A) compared to transitions within communities (green in Fig 4A), shows a positive relationship with Modularity (y-axis), defined as the ratio of the state prediction error elicited by between-wing transitions over within-wing transitions (corrected for "preferred" transitions; see Methods). (**B**) Between-community response time slowing (top) and modularity (bottom) show a positive trend and significant positive relationship with acquired total reward bonus respectively. (**C**) Between-community response time slowing (top) and modularity (bottom) show a positive relationship with the explicit hierarchical "rotation" choice policy. (**D**) Between-community response time slowing (top) and modularity (bottom) show a positive relationship with action-switching immediately upon leaving the goal wing ("Leave"). (**E**) Between-community response time slowing (top) and modularity (bottom) do not show a relationship with action-repetition when entering the goal wing ("Entry"). (**F**) Between-community response time slowing (top) and modularity (bottom) show a negative relationship with distance between paintings of the same community ("Within") (i.e., the paintings are placed closer together). (**G**) Between-community response time slowing (top) and modularity (bottom) show a positive relationship with distance between paintings at the boundaries of the same community ("Boundary"). (**H**) Between-community response time slowing (top) and modularity (bottom) show a negative relationship with distance between paintings connected across communities ("Bridge"). Significance of regression models indicated as *: $p < 0.05$, **: $p < 0.01$, ***: $p < 0.001$. For RT slowing, this significance corresponds to a simple linear regression. For SR modularity, the significance corresponds to a model comparison of a multiple regression model (including both RT slowing and SR modularity as independent variables) against a simple linear regression (including only RT slowing as independent variable).

were more affected by the community structure also reflected this community structure more strongly in their choice behavior. Including the modularity of the successor representation as a second independent variable leads to a significant improvement in model fit ($\beta = 0.768$, $F(2,116) = 10.694$, $R^2 = 0.156$, $\Delta F(1,116) = 6.443$, $p = 0.037$; Fig 8C, bottom). In addition, the data shows a positive relationship between response time slowing and the degree of response-switching when participants left the goal wing ($\beta = 0.003$, $F(1,117) = 14.642$, $R^2 = 0.111$,

$p = 0.002$; Fig 8D, top). Including the modularity of the successor representation as a second independent variable leads to a significant improvement in model fit ($\beta = 0.995$, $F(2,116) = 11.810$, $R^2 = 0.169$, $\Delta F(1,116) = 8.090$, $p = 0.032$; Fig 8D, bottom). By contrast, no relationship is observed between response time slowing and response switching upon goal entry of the goal wing ($\beta = 0.000$, $F(1,117) = 0.061$, $R^2 = 0.001$, $p = 1$; Fig 8E, top). Including the modularity of the successor representation does not yield any improvement in model fit ($\beta = 0.005$, $F(2,116) = 0.031$, $R^2 = 0.001$, $\Delta F(1,116) = 0.000$, $p = 0.979$; Fig 8E, bottom).

Although response times and choices reflect independent dimensions of the museum task, they are derived from the same behavioral unit (i.e. "the choice"). By contrast, the post-test free sorting task is completely independent from the behavior used to estimate modularity. Nevertheless, the data show a significant relationship between response time slowing and how close participants would sort paintings from adjacent rooms in the same wing ($\beta = -0.002$, $F(1,116) = 18.926$, $R^2 = 0.140$, $p < 0.001$; Fig 8F, top). Including the modularity of the successor representation as a second independent variable leads to a significant improvement in model fit ($\beta = -0.991$, $F(2,115) = 26.401$, $R^2 = 0.315$, $\Delta F(1,115) = 29.264$, $p < 10^{-5}$; Fig 8F, bottom). Interestingly, the data also showed a relationship between response time slowing and how far participants placed rooms that lie at the boundaries of the same wing, and are thus not connected ($\beta = 0.001$, $F(1,116) = 16.069$, $R^2 = 0.122$, $p < 0.001$; Fig 8G, top). Including the modularity of the successor representation as a second independent variable leads to a significant improvement in model fit ($\beta = 0.377$, $F(2,115) = 26.851$, $R^2 = 0.318$, $\Delta F(1,115) = 33.177$, $p < 10^{-5}$; Fig 8F, bottom). Higher values of both between-community slowing and successor representation modularity thus correspond to participants who correctly placed these rooms further apart, which in fact provides evidence against community structure and in favor of them accurately learning the true adjacencies. Similarly, the data showed a relationship between response time slowing and how closely participants would place two rooms that are connected between different wings ($\beta = -0.002$, $F(1,116) = 17.548$, $R^2 = 0.131$, $p < 0.001$; Fig 8H, top). Including the modularity of the successor representation as a second independent variable leads to a significant improvement in model fit ($\beta = -0.872$, $F(2,115) = 28.219$, $R^2 = 0.329$, $\Delta F(1,115) = 33.912$, $p < 10^{-6}$; Fig 8H, bottom). Higher values of both between-community slowing and successor representation modularity thus correspond to participants who correctly placed these rooms closer together, which is also in line with more accurate learning of true adjacencies. Collectively, these results indicate that participants who learned community structure using successor representations (as evidenced in response times), were better able to reconstruct the museum layout later from memory, while doing so with *less* bias towards reconstructing the underlying community structure.

## Discussion

Human behavior in the real world is characterized by long-range dependencies between action plans and outcomes [5,6]. Much of this information is high-dimensional and complex, but can be compressed to yield more efficient, generalizable representations [9,10,45–48]. Although how humans learn new behaviors according to such principles has remained elusive, previous work has shown that the computational burden involved in naturalistic behavior can be alleviated by setting "subgoals" that constrain planning over low-level actions [11,13–15,37,49]. It has been suggested that predictive representations such as the successor representation can be leveraged to identify such abstractions [17,23]. Predictive learning is ubiquitous in humans [50–53] and individual differences in predictive learning have previously been linked to individual differences in goal-directed decision making [54]. Specific evidence exists that humans can learn predictive representations that are sensitive to higher-order aspects of tasks

[29,30,40,41]. As far as we are aware, this study is the first to show that individual differences in higher-order predictive learning ability predict individual differences in hierarchical abstraction during goal-directed decision making.

In order to link predictive representation learning to hierarchically-informed decision making, we adapted a task that has been previously leveraged to show higher-order influences on predictive learning [30]. The museum task requires participants to navigate through a virtual museum, where the connections between rooms were characterized by community structure. Crucially, a navigational strategy based on a learned successor representation should differ from a strategy based on abstract, hierarchically structured representations, the latter of which are argued to be less cognitively demanding and faster to execute, even if otherwise sub-optimal [6,37,55–58]. This allows for investigating the emergence of hierarchical abstraction on a single task, and establishes a clear link with earlier work on predictive learning [23,29,59]. We find participants follow such a hierarchically abstracted strategy, consistently selecting the action that is guaranteed to reach the desired community (subgoal) and dynamically switching their behavior immediately following undesired transitions out of this community.

It has been shown participants are generally slower to respond to more surprising events [29,60,61]. Crucially, events that are part of separate communities are experienced as more surprising, even though these events are not in fact less predictable [29,40,41]. Notably, we replicated this effect in our novel goal-directed decision making task, and observed it to be significantly related to the modularity associated with the successor representation model.

Although previous studies on predictive learning did not investigate goal-directed decision making, recent work has shown that response times are influenced by the values of the different options in a value-based decision making task [32–36]. Note that these studies investigate decisions between stimuli leading to immediate outcomes instead of sequential decisions that require planning. Interestingly, it has been observed that learned values can invigorate the responses of rodents engaged in temporally extended behaviors, including by increasing their speed [62–64]. In particular, it has been argued that gradually increasing vigor when approaching a goal (i.e., a "goal gradient") [65,66] may be a core feature of flexible, goal-directed behavior [67–70]. We also observed a gradient in the response times in our task, indicating the use of a learned cognitive map to flexibly compute values for changing goals. However, the participants slowed rather than sped up when they approached the goal. We posit that this occurred because the task encouraged participants to slow their responses when nearing the goal location in order to identify the target painting and obtain the financial reward. By contrast, unlike higher positive values (which predicted slower responding), higher positive *reward prediction errors* predicted faster responding. However, our data suggest that instead of a relationship between response time and signed reward prediction errors (i.e., that distinguish between positive and negative outcomes), participants might instead slow down for larger absolute reward prediction errors (i.e., that collapse across the valence of the outcome). Exploratory modeling indeed suggested this relationship held more strongly (see S2 Appendix). This suggests that the effect of reward prediction error on response times is most parsimoniously explained as surprise-based slowing with respect to task progress.

Crucially, the museum task required participants to revise their value predictions in a flexible manner, as the goal locations were constantly changing. Tasks that involve changing goal locations cannot be simulated using "model-free" algorithms that have been previously leveraged to investigate the relationship between response times and value predictions [32–36]. Although this process is consistent with a canonical "model-based" reinforcement learning algorithm, model-based algorithms do not make specific predictions about the observed influence of community structure on response times. By contrast, a computationally simpler but less flexible successor representation algorithm [18,25] can do all of this. Here, successor

representation learning specifically accounted for the hierarchical abstraction observed in our choice data [23]. Further, the degree to which individual participants learned the community structure through the successor representation correlated with how effectively they applied a hierarchically abstract choice policy, although we note that this result is derived from an exploratory (i.e. not pre-registered) analysis. This pattern of results agrees with a theoretical model wherein participants learn a predictive representation of the task by engaging with it (as revealed by the successor representation model), and apply a secondary computational operation to this predictive representation in order to find an effective hierarchical representation useful for choice behavior (as revealed by the explicit hierarchical model).

Several algorithms have been proposed that implement this secondary computation by taking an adjacency matrix as (part of) their input [13,15,16,23]. This work generally assumes that the adjacency matrix has been accurately learned by the participants without considering how the specifics of the primary learning process could influence the abstraction process. Instead, we show that individual differences in learning a successor representation correlate directly with the efficacy of hierarchical abstraction. This finding is contrary to algorithms that rely on spectral graph theory wherein the relevant eigenvectors are constant over the entire range of possible discount factors [23], thus predicting no relationship between the estimated discount factors and the effectiveness of the hierarchical abstraction. Further, if abstractions rely on graph-theoretic measures of betweenness-centrality as described elsewhere [16], then the abstractions could be instantly computed from the successor representation as the "simple randomized shortest paths" betweenness, where the "fundamental matrix" corresponds directly to the successor representation [71] (algorithm 1). However, the quality of this abstraction increases with smaller discount factors, which is the opposite of what we find. Additionally, subgoal identification through betweenness-centrality does not explicitly assign lower-level states to higher-level state abstractions, leaving open the question why people would apply a simplified general "rotational" policy to all states in one community.

By contrast, our findings are more consistent with a Bayesian model of hierarchical abstraction as proposed by [15]. This model considers several factors for identifying efficient hierarchical representations, including dense connectivity within state abstractions, and sparse connectivity between them. When relying on a successor representation as input, as opposed to a ground-truth adjacency matrix, this density metric is by definition maximized when discount factors are most sharply tuned to capture the modularity of the environment. Similarly, (13, equation 7) propose an algorithm for computing the "entropic centrality" of different nodes. This computation can be performed directly by using the successor representation—which directly corresponds to the "fundamental tensor"—and substituting the degree of each node by the summed entries in the successor representation corresponding to that node. They observe that normative hierarchical abstractions often minimized differences in entropic centrality within state abstractions, and maximized this between state abstractions. This difference should be most sharply tuned in a successor representation when the discount factor emphasizes the modular structure most strongly, in line with our reported results.

Why would participants need to develop high-level (hierarchical) representations when the low-level successor representations already get the job done? Notably, both [13] and [15] explicitly assign lower-order states to higher-order state abstractions, and run separate planning algorithms at both levels. In each of these models, the lower-level planner should in principle be able to find policies with more fine-grained detail than following the simplified "rotational" policy. However, if we assume running the lower-level planner is computationally [72] (or cognitively, [6,56]) costly, agents might learn to omit running this planner, specifically when they can also learn that always executing the same action for every state in one state abstraction reliably yields good results (as in the Museum task). Although it has previously

been argued that policies can be directly computed by multiplying the successor representation with a reward vector [24], which should not be computationally expensive, in more complex domains or for longer timescales, generative sampling may still be required to reliably asses the outcomes of the possible actions on a single trial based on a learned successor representation [73–75]. Note that we only observe evidence for value computations based on a successor representation in the response times by fitting the model across many trials and many participants. Sampling from the successor representation would likely incur computational or cognitive costs e.g. [76], which could be alleviated by hierarchical abstraction. We speculate that participants follow the rotational policy in the Museum task mainly for this reason, despite the fact that their response times reveal sensitivity to relatively detailed representations of the underlying task structure. That participants' choice behavior can access such details is also illustrated by the significant decrease in the proportion of rotation choices when the rotation is in fact suboptimal (Fig 2F–2H).

Note that the particular successor representation model implemented here is relatively simple, as more advanced methods that yield better generalization in the face of changing goals have been suggested [77,78]. For example, [79] proposes a stable successor-like "default" representation that (typically) corresponds to a random policy. This representation can instantaneously adapt to "skew" with respect to the appropriate policy for a newly presented goal, unlike the typical successor representation which is updated through experience and thus skews towards recently pursued (but no longer relevant) goals. Some evidence for more advanced forms of successor-based generalization has been observed in human choice behavior [80]. Other research has shown that graph-topological features such as community structure can be inferred based on prior learning [81]. The latter research has also shown that this inference can lead to efficient navigation between task states by selecting the boundary nodes when navigating between communities. However, this is not evidence for a simplified hierarchical representation of the task, because any non-hierarchical planning mechanism also makes this same prediction. More generally, many prior studies investigating learning and generalization only analyze choice behavior. Instead, the current study illustrates how examination of response times can reveal signatures of predictive learning that may underlie more advanced forms of generalization in goal-directed learning.

In order to capture sensitivity to the probabilistic action-outcome contingencies in the task design, we modeled participant behavior using successor representations that predicted state-action conjunctions [24,26]. By contrast, the original formulation of the successor representation, later invoked as a model of hippocampal function, only predicts relationships between task states [18,23]. The state-state formulation is well-suited for simulating decision making in spatial contexts where the outcomes of specific actions are fixed (e.g., the action "go left" always results in moving left). Interestingly, the prediction of state-action-outcome conjunctions has been invoked as an explanation for several effects related to cognitive control and flexible decision making in the medial prefrontal cortex [82–86]. Additionally, recent evidence suggests that distinct dopaminergic populations may encode the prediction error associated with executing specific actions (although not their actual identity) [87–90]. The museum task was not specifically designed to disambiguate between predictive representations of states or state-action conjunctions, and although we observed a numerical preference for a state-action predictive model, we did not find any statistically reliable differences between these models (see S2 Appendix). When inspecting posterior predictions of response times for various rooms within a single community (Fig 5), we found that the state-action successor representation does not predict slowing in the boundary room that allows for a transition between two non-goal communities (blue). Because participants are prone to select the action that allows for rotation towards the goal community, they typically select the action allowing for a transition

between communities when in the boundary room that allows for a transition toward the goal community (orange). Conversely, participants are not very likely to select the action allowing for a transition between communities when occupying the boundary room that allows for a transition between two non-goal communities (blue). This implies a smaller state prediction error for the latter rooms for the state-action successor representation as compared to the state-state successor representation, because outcomes that transition between communities are excluded in the former but not the latter case. As a consequence, specifically the state-action model incorrectly predicts less slowing for rooms at the boundary of two non-goal communities. Additionally, the boundary room that allows for a transition towards the goal community is assigned a substantially higher expected value than the other two types of rooms, further contributing to the model-predicted slower response times.

Besides various implementations of the successor representation, our data could be explained by alternative sequence learning mechanisms, for example "free energy minimization" [29] or (cloned) hidden Markov models [91], in so far as these can implement a mechanism that is isomorphic to the discount factor of the successor representation. Note that the prediction of action (sequences) can form an alternative basis for hierarchical reinforcement learning [57,92–94], but this implies inflexible "open-loop" control. The museum task contained probabilistic action-outcome mappings and required participants to search for a new goal location on every miniblock, preventing against this strategy. The interaction between action sequences and state based hierarchical learning could be an interesting direction for future research.

Note also that the successor representation yields a vector error signal that indicates the degree to which predictions of multiple possible upcoming events need to be adjusted. This computation agrees with the distributed nature of the prediction error signals observed in the medial prefrontal cortex [82,84,86,95] and in the midbrain dopamine system [20,96]. However, modeling of surprise-based slowing of response times requires that this error vector be converted to a scalar signal. Toward this end, previous work has modeled response times using a successor representation by normalizing the successor matrix to resemble a transition probability matrix. The scalar surprise signal was then obtained by looking at the transition probability from the original to the outcome state [97]. In line with the intuition that the successor representation estimates the "expected future discounted state occupations", this work specified the target for the update as a one-hot vector of the outcome state [44,97]. However, it has been noted that the correct target for the update is a one-hot vector of the original state summed with the discounted successor vector of the outcome state [18,24]. Hence, the surprise signal should reflect this latter component, which is vectorial. For this reason, we modeled the response times with a formal distance metric (the angular distance) in order to yield a scalar surprise signal.

Not only were the individuals who showed greater sensitivity to community structure in their response times (as measured through the discount factor of the successor representation) more sensitive to hierarchical abstractions in their choice behavior, they also reconstructed the lower-order structure of the museum more accurately from memory. This included more accurate separation of the "boundary" rooms within the same community, which counterintuitively indicates less bias with respect to community structure. We speculate that the individual differences in the estimation of the discount factor reflect how engaged the participants were with the task. In particular, we suggest that individuals who paid more attention to the task learned more accurate successor representations of the environment, which simultaneously provided more information about both low-level (room) and high-level (wing) task structure. Interestingly, [29] proposed that the learning of community structure is mediated by "mental errors", which should be more prevalent for participants with lower task engagement. Our

data instead suggest that the learning of community structure improves with higher task engagement.

In addition to the behavioral evidence for successor representation learning [26,28], functional magnetic resonance imaging of humans performing incidental learning tasks has revealed successor-like representations in the hippocampus [98–101]. During decision making behavior, the medial frontal cortex and specifically the anterior cingulate cortex show sensitivity to hierarchical features of tasks [102–104]. Our results can be interpreted within this context, where the hippocampus might be involved in general statistical learning [105], whereas the anterior cingulate cortex represents variables relevant for decision making, specifically models that can be leveraged for hierarchical reinforcement learning [6,106]. These hierarchical models may be extracted from successor-like hippocampal representations as a form of "schema" learning or meta-learning [107–109]. Future work can leverage neuroimaging techniques to search for task representations and prediction errors associated with successor learning and hierarchical reasoning in the museum task, and whether these are represented in distinct neural regions, potentially following a rostro-caudal abstraction gradient associated with frontal cortical function [110].

## Methods

### Museum tour guide game

A preregistration of the museum task and our main analyses can be found at https://osf.io/n2jcz/, including all of the relevant code and data. The task was programmed with JsPsych version 7.1.0 [111] using stimuli provided by [112]. Data for 141 participants were collected on the Prolific platform [38]. All participants gave their informed consent prior to participation and the study was conducted according to the guidelines of the General Ethical Protocol of the Faculty of Psychology and Educational Sciences (Ghent University) and the ethical standards prescribed in the 1964 Declaration of Helsinki. The data of 21 of participants were excluded based on preregistered criteria: 3 of these participants chose the same action on over 90% of the trials, and 18 of these participants selected the wrong goal or missed the correct goal on over half the miniblocks, both of which are indicative of task disengagement. Data of the free sorting task was corrupted for 1 participant, meaning this participant was not included in the analysis of the free sort distances (but still included in all other analyses).

### Training and testing phase

We wanted participants to learn the structure of the environment from experience, including the three communities or "wings" of the museum. The learning process can be captured with a computational model as described below. Importantly, this means participants were not instructed on any aspect of the layout of the rooms of the museum, so as not to bias their learning and decision making behavior prior to task experience. Instead, we designed a training phase where participants would see every goal painting and possible distance between start and goal location exactly the same number of times. This ensures an unbiased predictive representation of the museum can be learned from experience. We designed a separate testing phase where we could then investigate the final decision making behavior of the participants.

In the training phase, the participants were given exactly 75 goal paintings to search for (i.e. 75 "miniblocks"). They were provided an opportunity for a short rest every 25 miniblocks. By contrast, the test phase did not limit the number of miniblocks to perform. Instead, the participants were given a "budget" of 1000 room transitions where each room they visited deducted 1 "step" from this budget. Importantly, efficient policies enabled participants to visit more goals in a limited number of steps and thus accrue more reward (see S1 Appendix). The participants

were given the opportunity for a short rest approximately midway through the test phase, namely after completing the miniblock when their budget dropped below 500 steps. The testing phase terminated after the miniblock on which their budget dropped below 0 steps. The reward for finding each goal painting in the training phase was £0.10. This increased to £0.15 in each testing phase. The penalty for indicating the wrong goal or missing the correct goal was always £0.02.

In the training phase, combinations of start and goal locations were exactly balanced, preventing against the development of model-free preferences for specific rooms (see S1 Appendix). In the testing phase, the start and goal locations, and the distances between them, were no longer balanced on the assumption that the participants learned an unbiased representation of the museum layout in the training phase. In the testing phase, the goal locations were sampled with a uniform probability from the 10 rooms outside the starting wing. The previous goal location was always the next starting location, just as in the training phase.

## Action-outcome mapping

Participants were requested through Prolific to perform the task using a QWERTY-layout keyboard. On each trial, they were required to select between key <z> and key <m> on their personal keyboard. For each state, each key mapped at random with equal probability to two possible outcome states (Fig 1B). Which key mapped to which set of outcomes was determined randomly for each participant, but consistent across wings, meaning there were two possible action-outcome mappings. A primary feature of this action-outcome mapping was that each key consistently mapped the participant out of the current wing in only one direction (labeled as "intention" in Fig 1B).

This consistent "rotational" mapping ensured that the participants could exercise direct control over movement between the museum wings, but not between the individual rooms. In fact, the action-outcome contingencies were specifically designed to be policy independent, such that irrespective of whether the participant behaved completely randomly or they consistently selected the same key over the entire course of the experiment, the expected number of times each room was visited was equated for all of the rooms (see S1 Appendix), as implemented previously in tasks relying on random walks [29,30,40,41]. In order to maintain the possibility to dissociate hierarchically organized behavior from detailed model based behavior, this required constraining the action-outcome contingencies by replacing 4 specific directed (action-dependent) transitions with a transition that was also available for the other action (Fig 1C). Extensive justification for these constraints, which were controlled for statistically, are provided in S1 Appendix. Fig 9 illustrates further how these constraints affect the transition types within a wing. In our analyses, we included binary regressors coded as 1 for the "preferred" transitions (orange), which are common to both <z> and <m> choices from the outgoing room, and for the "wide node" transitions (yellow), which end in the only room that does not have preferred outgoing transitions (and is thus the only room that has 4 possible direct outcomes instead of just 3). Finally, we included a binary regressor coded as 1 for "wide trans" (pink), which are less predictable outgoing transitions from this wide node.

## Computational modeling

As preregistered, miniblocks that contained a "goal miss" or "wrong goal" trial (see Fig 1D) were excluded from the model fits. Computational models for the response times and choices were estimated with Stan [113] using Hamiltonian Monte Carlo (HMC). All models had no divergences and converged appropriately to a Gelman-Rubin $\hat{R}$-statistic lower than 1.01. For each model for each participant we computed the approximate leave-one-out cross-validation

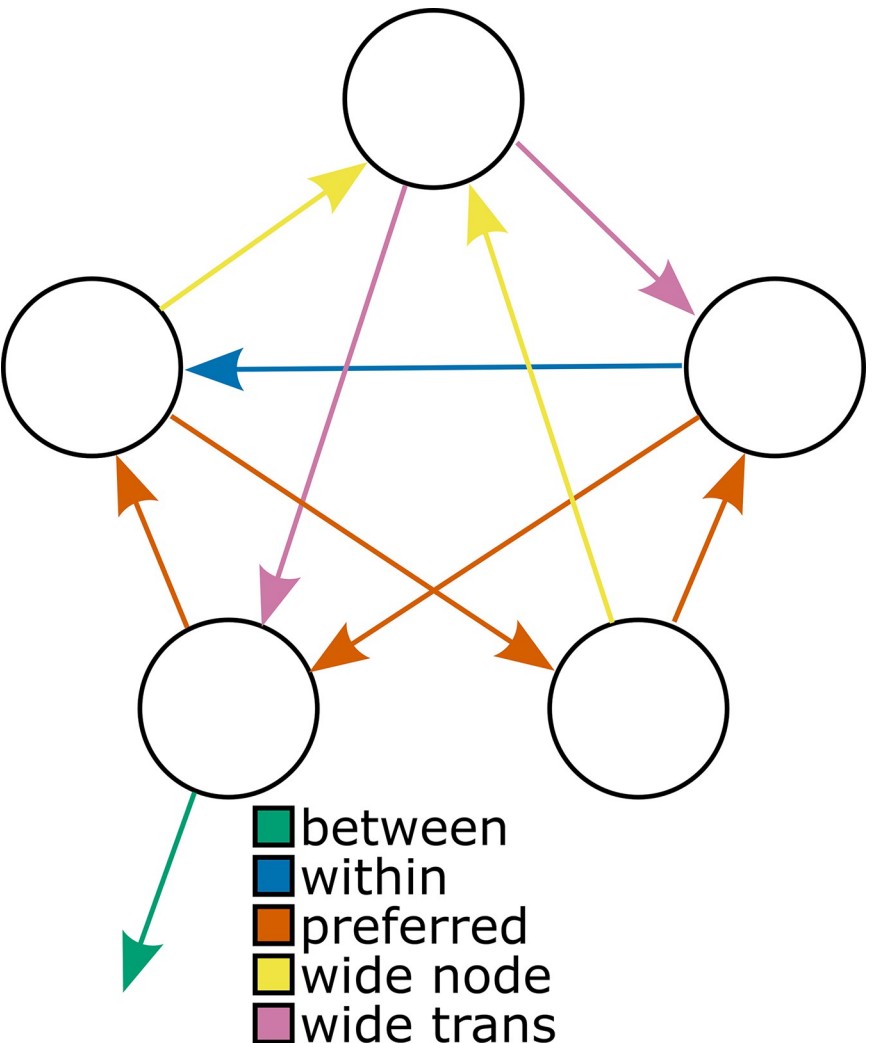

**Fig 9. Transition types with respect to preferred transitions.** Arrows colored by transition type with respect to the preferred and removed transitions as shown in Fig 1C. Here, preferred transitions are coded as orange ("preferred") (blue arrows in Fig 1C). These can be the outcomes of selecting both <z> and <m> in their outgoing states. These transitions were included as nuisance regressors in our models, since we expect them to yield faster response times as they are more predictable. It can be seen that only the top node does not have any outgoing preferred transitions, and when referencing Fig 1B, it can be seen this is the only node with 4 possible immediate outcomes (2 distinct outcomes for each action). Hence, this node is referred to as "wide node" and transitions into this node are labeled as such (yellow). Transitions out of this node are labeled as "wide trans" (pink). Both these transitions can be expected to yield slower response times, since the more diffuse nature of transitions out of this node makes them harder to anticipate. Therefore, both these transition types are included as nuisance regressors in our models. It can be seen only one within-wing transition is not preferred and not associated with the wide node (blue, "within"). This transition can be contrasted with the "between" wing transition (green) to yield an unbiased estimate of higher-order surprise-based slowing.

score using Pareto Smoothed Importance Sampling (PSIS-LOO) [114]. PSIS-LOO has been shown to penalize complex models more appropriately than AIC and WAIC [115]. PSIS-LOO yields an estimation for the "expected log pointwise predictive density" $\hat{elpd}^k_{loo}$ for each model $k$ for every participant. We use this to compute the relative evidence for each model for every

participant as Akaike-like exponential weights $w_k = \frac{\exp(\hat{elpd}^k_{loo})}{\sum_{k=1}^{K} \exp(\hat{elpd}^k_{loo})}$ and correct them for

estimation uncertainty, regularizing them further away from 0 and 1, using the Bayesian bootstrap [116]. This is easily calculated from HMC chains sampled with Stan and referred to as *Pseudo-BMA+* weighting. Given enough data, Pseudo-BMA+ weighting should converge to 1 for the model that most closely resembles the true data generating process. It is therefore considered an $\mathcal{M}$-closed model comparison procedure. It is important to validate that the winning model is not only better than the other considered models, but also describes the data adequately in the first place [117]. In order to show this, we used posterior predictive simulations of the winning model and compared these to the real data.

Participant-level model evidences were submitted to random-effects group-level Bayesian model selection (GroupBMS) [118]. This allowed us to compute a Bayesian omnibus risk (*BOR*) value, which indicates whether any model among a set of models is more or less prevalent in the population than others. It also allowed us to compute a protected exceedance probability (*pxp*) for each model, which is an estimate of how likely that model is to be the most prevalent. Lastly, it allowed us to get a participant-level estimate of the posterior model probability, indicating how likely the data from each participant are to be generated by each of the considered computational models.

### Response time model specification

Response times were modeled as a regression on the mean parameter of a shifted log-normal distribution. We fitted 4 chains with 1000 warmup and 2000 post-warmup samples each, for a total of 8000 post-warmup samples. We used a treedepth of 13 and a warmup-acceptance of 0.999. The structure of the model is

$$
\begin{aligned}
(y_{ij} - ndt_{im}) &\sim lognormal(\alpha_{im} + \boldsymbol{\beta}_{im}\boldsymbol{x}_{im}, \sigma_{im}) \\
\alpha_{im} &\sim \mathcal{N}(6, 1.5) \\
\boldsymbol{\beta}_{im} &\sim \mathcal{N}(0, 0.1) \\
\sigma_{im} &\sim half\mathcal{N}(0, 1) \\
\gamma_{im} &\sim Beta(1, 1) \\
ndt_{im} &\sim \mathcal{U}(0, u_i)
\end{aligned}
$$

Where $i$ indexes the relevant participant, $j$ the relevant trial, and $m$ the relevant computational model. $y$ represents the response time and $\alpha$ represents and intercept. $\boldsymbol{\beta}$ represents a vector of regression slopes and $\boldsymbol{x}$ a vector of independent variables. $ndt$ represents the non-decision time with $u_i$ representing the minimum response time for participant $i$, and $\sigma$ represents a variance parameter of the log-normal distribution. Note that the discount factor $\gamma$ is only part of the model-based and the successor representation model, where it is used respectively to discount expected value directly, or the expected future state occupations, which are entered as independent variables into the model (see below) hence allowing for the estimation of the discount factor.

*Model-based model.* In order to compute the expected value for the model-based agent, we precomputed a value iteration [3] until convergence for each of the 15 possible rooms as the goal room. The discount factor is part of the value iteration algorithm. Since we estimate the discount factor, the value iteration was performed within the Stan program, again for every new sample Stan would take. We assumed a completely accurate transition model of the environment and a reward vector **r** where every entry was a small cost value (-0.08) except for the relevant goal room, which was set to 1. This parameterization of the cost value is irrelevant for our modeling, as all values below 0 yield exactly collinear regressors [119].

In the regression model, we considered specifically the EV of the action that was selected on that trial. The reward prediction error (RPE) was then computed as the EV of the current room minus the EV of the previous room. The conflict was computed as the absolute difference in EV between the two available actions and referred to as $EV_{diff}$. Since both RPE and $EV_{diff}$ were derived from EV, there was moderate correlation between these regressors. To control for this, we successively orthogonalized them using the Gram-Schmidt process before fitting the model. The EV regressor was left intact. The RPE regressor was orthogonalized with respect to the EV regressor $RPE_{\perp} = RPE - \frac{<EV,RPE>}{<EV,EV>} EV$ where $<v_1, v_2>$ denotes the dot product of two vectors. The $EV_{diff}$ regressor was then orthogonalized with respect to both the EV and the orthogonalized RPE regressors $EV_{diff\perp} = EV_{diff} - \frac{<EV,EV_{diff}>}{<EV,EV>} EV - \frac{<RPE_{\perp},EV_{diff}>}{<RPE_{\perp},RPE_{\perp}>} RPE_{\perp}$. Regressors were standardized after orthogonalization. This way, each regressor only captured unique variance associated with it. Significant parameter estimates could then be directly interpreted as evidence that participants are sensitive to that independent variable. Note that shared variance is attributed to the regressor that comes "earlier" in the order of orthogonalization. Since these independent variables are dependent on the discount factor and the value iteration, they were recomputed for every new sample Stan would take. Orthogonalization and standardization thus happened within the Stan program.

*Successor representation model.* The successor representation agent learns a matrix **M** of expected future (discounted) state visits. If there are *A* different actions and *S* different states in the environment, $\mathbf{M} \in \mathbb{R}^{|SA| \times |SA|}$. Each entry $\mathbf{M}_{sa,sa'} = \mathbb{E}[\sum_{t=0}^{\infty} \gamma^t \mathbb{I}_{sa=sa'}|s_0 = s, a_0 = a]$ where *sa* indicates the previous state-action pair, and *sa'* indicated the next state-action pair. This can be learned following a SARSA update rule. Starting **M** out as an identity matrix for each state-action pair, updates follow:

$$\mathbf{M}[sa_t, :] \leftarrow \mathbf{M}[sa_t, :] + \lambda(1_{sa_t} + \gamma\mathbf{M}[sa_{t+1}, :] - \mathbf{M}[sa_t, :])$$

where $1_i$ indicates a one-hot vector with a 1 at index *i*. The SR introduces two parameters to estimate: The discount factor $\gamma$ and the learning rate $\lambda$. For all our model fitting, we kept the learning rate $\lambda$ fixed at 0.1 [44,97].

The state prediction error (SPE) regressor for trial *t* was computed as the angular distance $\frac{1}{\pi} \arccos\left(\frac{v_1 \cdot v_2}{||v_1||||v_2||}\right)$ between $\mathbf{M}[sa_t,:]$ and $\mathbf{M}[sa_{t-1},:]$. The angular distance is a formal distance metric computed from the cosine similarity of two vectors. An angular distance metric is useful in this case, as it is not sensitive to the magnitude of the successor representation vectors, which grow larger for larger discount factors, but only to the relative difference in which states are predicted more or less. We computed the EV as $\mathbf{M}[sa_t,:]$ **r** to model proximity to the goal, where **r** is defined similarly as for the model-based agent. RPE was the difference between EV of the current trial and the previous trial. Conflict was the difference between EV for the two different available actions. Orthogonalization similar to the model-based agent was applied to the EV, RPE, and conflict regressors, after which all regressors were standardized. Note that the discount factor was part of the Stan program, and thus **M** and the regressors derived from it are different for each new sample of $\gamma$ that Stan would take. Orthogonalization and standardization thus happened within the Stan program for each new sample Stan would take.

*Population-level estimate of parameters.* Assuming there is a clear winner in the model comparison procedure (i.e *BOR* < 0.05), we wanted to test whether a particular independent variable of that model *m* had a significant effect on the response time in the population. It is reasonable to pool information across participants and fit a hierarchical model for the winning model *m* if there is strong agreement on the optimal model in the sample [120]. A hierarchical

model fits all participants simultaneously and combines information across participants to acquire more accurate parameter estimates. However, the successor representation model is too computationally demanding to fit simultaneously across all participants. For this reason, we opted to first fit the models separately to each participant. We then combined the single-participant estimates of the regression slopes into a population distribution and tested if this was significantly different from 0. Note that this is not identical to fully hierarchically estimating the model: Different parameters cannot constrain each other in this approach (e.g. allowing for population-level influences on the individual regression slopes might affect the estimation of the discount factors, but this cannot be captured by our current approach). Separately fitting models to each participant and then combining their estimates to test for an effect is also called "two-stage regression" [121], and is also common in e.g. functional magnetic resonance imaging (fMRI) analysis [122].

Instead of testing point-estimates against 0 as typical in maximum-likelihood two-stage regression, we wanted to maintain participant-level information of the width of the posterior distribution, acknowledging the amount of uncertainty for each participant. We implemented a model analogous to the one suggested by [123], where we summarized the participant-level posterior distributions for each regression slope with a Gaussian parametric density estimation of the HMC samples. While this can omit information about co-dependencies between parameters [124], it is not expected to yield false positive results, since these (unmodeled) co-dependencies should widen the marginal posterior distributions, effectively reducing our power. We computed posterior means $\hat{\boldsymbol{\beta}}_i$, and posterior standard deviations $\boldsymbol{\varepsilon}_i$ for each participant $i$, which are $O$-dimensional vectors with an entry $o$ for each independent variable. We wanted to estimate the population-level parameters $\boldsymbol{\mu}$ and $\Sigma$, where $\boldsymbol{\mu}$ is an $O$-dimensional vector of population means $\mu_o$ for each independent variable, and $\Sigma$ is an $O \times O$ covariance matrix. We fitted 4 chains with 2000 warmup samples and 20000 post-warmup samples each. We used a treedepth of 13 and a warmup-acceptance of 0.999. The structure of the model is

$$\boldsymbol{\mu} \sim \mathcal{N}(0, 0.1)$$
$$\Sigma \sim \mathrm{diag}(\boldsymbol{\tau}) \cdot \boldsymbol{\Omega} \cdot diag(\boldsymbol{\tau})$$
$$\boldsymbol{\tau} \sim half\,\mathcal{N}(0, 0.1)$$
$$\boldsymbol{\Omega} \sim LKJ(1)$$
$$\boldsymbol{\beta}_i \sim \mathcal{N}(\boldsymbol{\mu}, \Sigma)$$
$$\hat{\boldsymbol{\beta}}_i \sim \mathcal{N}(\boldsymbol{\beta}_i, diag(\boldsymbol{\varepsilon}_i))$$

Where LKJ stands for the Lewandowski-Kurowicka-Joe distribution, a distribution over correlation matrix $\Omega$, and $\boldsymbol{\tau}$ is a vector of population standard deviations for the $O$ independent variables. The model gives us regularized estimates of participant level regression slopes $\boldsymbol{\beta}_i$ and, of main interest, population level effects $\boldsymbol{\mu}$. The latter were tested for significance as described in the section on Statistical analyses.

## Choice model specification

Choices were modeled based on a response-coded logistic regression, where key <z> was coded as "0" and key <m> is coded as "1". We only considered trials outside the goal wing, since this is where the explicit hierarchical model makes interpretable and differentiable predictions. We fitted 4 HMC chains with 1000 warmup and 2000 post-warmup samples each, for a total of 8000 post-warmup samples. We used a treedepth of 13 and a warmup-acceptance of

0.999. The structure of the model is

$$y_{ij} \sim bernoulli(\alpha_{im} + \beta_{im}x_{ijm})$$
$$\alpha_{im} \sim \mathcal{N}(0, 2)$$
$$\beta_{im} \sim \mathcal{N}(0, 2)$$
$$\gamma_{im} \sim Beta(1, 1)$$

Where $i$ indexes the participant, $j$ the relevant trial, and $m$ the relevant computational model. Note that the discount factor $\gamma$ is only part of the model-based and the successor representation model. We only analyzed choices outside the goal wing, since the hierarchical structure of the task only informs choices leading to the goal wing, and not inside that wing. We were specifically interested in whether participants use that hierarchical structure to guide their choices.

For the logistic regression, the null-model only contains an intercept, so in actuality there was no $\beta$ parameter for this model. It only models bias to select either <z> or <m>. The three cognitive models all contained one regressor (besides an intercept). For the model-based agent, we computed the expected value of the action <m> minus the value of action <z>, and standardized this within-participant. For the explicit hierarchical model, we computed an effect-coded regressor which was -0.5 when <z> was the current "correct rotation" and 0.5 when <m> was the current correct rotation. For the successor representation agent, we computed the expected value difference of <m> minus <z>, and standardized this within-participant.

*Population-level estimate of parameters.* In case one model clearly wins the model comparison (i.e. *BOR* < 0.05), we tested whether the relevant regression slopes differ from 0 in the population. We tested this hypothesis with a hierarchical logistic regression implemented in *brms* [125]. The prior for the population-level effect was left at the default, being a student-t distribution with 3 degrees of freedom, mean 0, and scale 2.5. The model included separate participant-level intercepts and slopes. We fitted 4 HMC chains each with 1000 warmup and 5000 post-warmup samples each, for a total of 20000 post-warmup samples.

## Posterior predictive checks

For the posterior predictions of choice and response switches (Fig 2) and of the response times (Figs 4 and 5), we draw one posterior prediction per trial for every sample accepted by HMC. In every case, this leads to 8000 posterior predictions per trial. For the response times, the posterior predictions were computed from the participant-level model estimation, again drawing one posterior prediction per trial for every accepted sample. For each trial, we took the mean of the 8000 posterior predictions as the mean posterior prediction for that trial. Separately, we assigned each trial regressor values based on the successor representation model, based on the discount factor of the maximum a-posteriori (MAP) estimate of the model. This means uncertainty about the discount factor is not taken into account in our posterior predictive check. This was necessary to overcome limitations in RAM.

## Statistical analyses

Where we mention the use of multiple comparison correction, used in the binary model comparisons and in the exploratory individual difference analysis, we multiplied the reported *BOR*- or *p*-values according to the Holm-Bonferroni procedure before reporting them [126]. For the significance testing of population-level parameters in hierarchical Bayesian linear models fitted for the winning choice and response time computational models, we report the

posterior mean ($M$), the 95% highest density interval ($HDI_{95\%}$), and the evidence ratio ($ER$) of every parameter [127]. $HDI_{95\%}$ corresponds to the smallest continuous interval that includes 95% of the total posterior density. If 0 is not included in this interval, this is suggestive of a meaningful effect of that independent variable with respect to the dependent variable. Additionally, we report the evidence ratio in the favored direction (for positive effects $ER_+$, for negative effects $ER_-$). In the case of $ER_+$, this is the ratio of the posterior density above 0 over the density below 0. $ER_-$ is the inverse of this. If 95% of the posterior density lies exclusively above or below 0, the corresponding $ER$ will be higher than 19. $ER$ will be infinitely large if all HMC samples lie either exclusively above or exclusively below 0. We consider an effect to be reliable when both the $HDI_{95\%}$ excludes 0, and the relevant $ER$ is larger than 19. The computation of the $ER$ and the definition of a cutoff value provide a 1:1 correspondence with frequentist p-values [128].

## Response switches

Response switching behavior was modeled with *brms* and the same prior structure as the population Choice model. The outcome variable was binary coded, with response repetitions coded as "0" and response switches indicating the participant responded different on trial $t$ than on trial *t-1* coded as "1", only considering trials that are part of the same miniblock. Entry and leave regressors were dummy coded with "1" indicating a trial immediately after the participant entered or left the goal wing. We also included a nuisance regressor that counted the (log-transformed) number of times an action has been repeated successively, which was standardized within-participant.

## Rotation and antirotation

We modeled the probability of selecting the correct rotation with *brms* and the same prior structure as the population Choice model. The outcome variable was binary coded, with selection of the correct rotation coded as "1" and selection of the antirotation coded as "0". A regressor corresponding to whether the correct rotation was currently <z> or <m> was added, coding <z> as -0.5 and <m> as 0.5. A second regressor was added, considering whether the current room implied the rotation was in fact optimal (blue, Fig 2F; coded as 0.5) or whether the antirotation was in fact optimal (orange, Fig 2F; coded as -0.5). We also included their interaction, to investigate whether the latter effect is especially present for either <z> or <m> rotations.

## Free sort

We modeled the Euclidean distances between the 105 pairs of paintings sorted by the participants using *brms* and the same prior structure as the population Choice model. First we standardized the Euclidean distances within participant, to correct for individual differences in participants who would use the available grid up to a different width (i.e. some place most paintings near the center, some place paintings all the way out in the corners). We dummy-coded three possible relationships between pairs of paintings. We defined a "community" regressor which was 1 when two paintings were part of the same community, and 0 otherwise. We also defined a "bridge" regressor which was 1 when two paintings were connected between two communities (i.e. not part of the same community) and 0 otherwise. Additionally, we defined a "boundary" regressor which was 1 for paintings that were part of the same community but not connected (i.e. the two boundary paintings within the same wing) and 0 otherwise. Note that for these paintings the "community" regressor is also 1, meaning this boundary regressor models whether participants place the boundary paintings further apart than other same-community paintings. Paintings that were neither part of the same community, nor had any direct connection between them, were captured by the intercept of the model. In order to

inspect bias induced by the community structure of the environment, we tested whether participants placed connected paintings from the same wing (community) closer together than connected paintings from different wings (bridge), by subtracting the "bridge" samples from the "community" samples and testing whether the new chain was significantly below 0. We ran an additional hierarchical Bayesian linear regression considering only pairs of paintings that had a shortest path length of 1 intermediate room between them. This allowed us to test whether the community-boundary paintings were placed closer together than other paintings with the same distance between them. This model consisted of only the "community" regressor and an intercept.

## Modularity measure

The successor representation can represent the community structure present in the task more or less strongly dependent on the value of the discount factor parameter. Our work differs from previous modeling work on community structure [29] in three important ways. (i) We do not standardize the successor matrix to a transition probability matrix after each new experience (see discussion), (ii) we model sequences of state-action conjunctions instead of states, and (iii) some states can succeed specific states more often than others ("preferred" transitions, Fig 9). We designed a measure analogous to the one reported by [29], but addresses the additional complexities of our task design. Our measure is based on the notion that, by definition of modularity, the successor representation prediction error should be higher for transitions between wings than for transitions within wings. However, since some transitions are "preferred" (see section "Action-outcome mapping"), these are expected to yield lower prediction errors, thus inflating our measure of community structure (see S1 Appendix). We thus compared specific non-preferred within-wing transitions ("within", blue; Fig 9) to between-wing transitions which are always non-preferred ("between", green; Fig 9). These transitions are topologically identical except for their distinction as between- or within-wing, and thus provide an unbiased estimate of the sensitivity to community structure. We defined the "modularity" measurement as the ratio between the mean angular distance of the successor representation vectors (the successor representation state prediction error) for all possible between-wing transitions and these non-preferred within-wing transitions. Note that this yields a "difference of distances" measurement analogous to representational similarity analysis techniques [129].

We computed the modularity measurement for each participant using the successor representation matrix after the testing phase. Note that we obtained a full posterior distribution over the discount factor $\gamma$, and thus over successor matrices. We reduced this to a point-estimate for the modularity by computing the expectation of modularity over the discount factor. To test for evidence of modularity at the population level, we first computed the expected modularity under the uniform prior distribution of the discount factor $\pi(\gamma){\sim}Beta(1,1)$. Because each participant made different choices and experienced a different sequence of states, there are variations in the relationship between the discount factor and the modularity (see Fig 7A and 7B), so this provides a proper "null hypothesis" for the modularity of each specific participant. We then computed the expected modularity under the recovered posterior distributions $p(\gamma|\gamma)$ for each participant. We compared these prior and posterior expectations using a paired-samples t-test.

## Supporting information

**S1 Appendix. Validations of task and computational models.** Discusses simulations of different agents on the task, details on model and parameter recovery, and details regarding

parameter estimation on the empirical data.
(DOCX)

**S2 Appendix. Control model fits.** Discusses several variants of the computational models and the robustness of the results under these variations.
(DOCX)

## Author Contributions

**Conceptualization:** Sven Wientjes, Clay B. Holroyd.

**Data curation:** Sven Wientjes.

**Formal analysis:** Sven Wientjes.

**Funding acquisition:** Sven Wientjes, Clay B. Holroyd.

**Investigation:** Sven Wientjes, Clay B. Holroyd.

**Methodology:** Sven Wientjes.

**Project administration:** Sven Wientjes.

**Resources:** Clay B. Holroyd.

**Software:** Sven Wientjes.

**Supervision:** Clay B. Holroyd.

**Validation:** Sven Wientjes.

**Visualization:** Sven Wientjes.

**Writing – original draft:** Sven Wientjes.

**Writing – review & editing:** Clay B. Holroyd.

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
