## [Decision Letter · Decision Letter 0]

27 Oct 2023

Dear Dhr. Wientjes,

Thank you very much for submitting your manuscript "The successor representation subserves hierarchical abstraction for goal-directed behavior" for consideration at PLOS Computational Biology.

As with all papers reviewed by the journal, your manuscript was reviewed by members of the editorial board and by several independent reviewers. In light of the reviews (below this email), we would like to invite the resubmission of a significantly-revised version that takes into account the reviewers' comments.

As you will se below, the reviewers raised major concerns regarding explanations of the relationship between models, as well as the choices that went into the modeling (regressors etc.). All of which influences the interpretations and conclusions. These may however be factors that are possible to address in a revision.

We cannot make any decision about publication until we have seen the revised manuscript and your response to the reviewers' comments. Your revised manuscript is also likely to be sent to reviewers for further evaluation.

Sincerely,

Ulrik R. Beierholm

Academic Editor

PLOS Computational Biology

Lyle Graham

Section Editor

PLOS Computational Biology

Reviewer's Responses to Questions

**Comments to the Authors:**

Reviewer #1: Summary:

The authors run a series of experiments to investigate how humans learn abstract representations with an underlying community structure in the context of goal-directed behavior. The authors find that participants’ choices reflect the community structure of the task and that their reaction times are best predicted by the successor representation.

Overall, I find the experiments and analyses interesting and convincing. Whereas previous graph-learning studies have mostly involved passive learning, the authors study goal-directed behaviors. Although I’m not convinced by the use of the word “hierarchical” (see below), I do believe that this study provides important insights into goal-directed behaviors in environments with community structures. Below I provide two specific comments for the authors to consider.

Comments:

The authors refer to “hierarchical” structure throughout the paper (and even in the title). In their experiments, they consider an environment with two levels of structure: the fine-scale structure of the connections between rooms and the large-scale structure of the wings (or tightly connected groups of rooms). Traditionally, this sort of two-level structure is referred to as “community” structure, as the authors correctly point out throughout the paper. In fact, this exact graph structure has been studied previously, always (to my knowledge) in the context of community structure. Hierarchical structure, by contrast, typically refers to nested communities with at least 3 levels of description (see a recent preprint on learning hierarchical structure: https://arxiv.org/abs/2309.02665). While this is simply a matter of semantics (not a critique of the research itself), in keeping with previous work this paper would be more accurately described as a study of community structure learning rather than hierarchical structure.

When defining the transition structure between rooms, the authors choose a graph with some directed transitions that are not allowed while their reverse occur twice as often as all other transitions (Fig. 1C). However, this choice is likely to have undesirable consequences for the behavior of the participants. For example, having some transitions that occur twice as often as the rest is likely to lead to differences in reaction times (which the authors then have to account for in their regressions). These differences in transitions then give rise to differences between nodes. For example, node 3 has four different incoming and outgoing transitions, while nodes 1, 2, 4, and 5 only have three different incoming and outgoing transitions. I believe the authors could have avoided these issues with a different choice of allowed transitions. For example, in Fig. 1Bii the authors could have allowed directed transitions (1,2), (1,3), (2,3), (2,4), (3,4), (3,5), (4,1), (4,5), (5,2), and (5,6). Then the transitions for the other response would just be the direct reverse. Can the authors explain why they don’t choose transitions such as these, which preserve the symmetries between different connections and different nodes? If the authors have a justification, this should be included in the main text. And in any case, the authors should discuss the potential impacts of these asymmetries on human behavior.

Reviewer #2: Summary

Wientjes and Holroyd present a study aiming to test how individuals compute abstract hierarchies over states for purposes of goal-directed behavior. Some recent work (Botvinick et al. 2014; Stachenfeld et al., 2017, Machado et al., 2023) has proposed that this can be done using the successor representation. The current manuscript presents human behavioral data from a task aiming to test a version of this theory. In the task, participants navigated between ‘rooms’ arranged as nodes on a graph. Nodes were arranged so that each node belonged to one of 3 communities, that were only interconnected by a single transition between two nodes. Each node had two actions, which instantiated probabilistic transitions to one of two other nodes. Actions were arranged so that one action tended to move one either clockwise or counterclockwise (with some exceptions) through nodes in a community.

In analyzing choices, the authors found evidence that participants used a strategy which reflects knowledge of the graph community structure. Specifically, their choices were best explained by a model (the explicit hierarchical model; EH) which chooses a single action for each cluster (specifically, chooses the parameter which rotates one in the best direction out of the cluster for the current goal). In contrast, response times were best explained by an SR model. Between participants, there was variability in the extent to which the fitted SR model’s predictions changed when moving between clusters (due to variation in discount factor) - a measure of SR capturing hierarchical structure (“modularity”). Between-participant variability in this modularity measure (derived from the SR model’s fit to response times) explained variance in a number of measures related to hierarchical structure in choices and also in a separate task where participants had to estimate the spatial position of each node.

It is argued that these results support the theory that individuals learn the SR from experience and then apply a secondary computation to the predictive representation to find a hierarchical representation for choice behavior.

Assessment

There is a lot that I like about this manuscript. How people identify hierarchy in temporally extended tasks is an important problem in cognitive science. That the SR is used for this is a compelling hypothesis. The authors preregistered their analysis and mark where in their analysis they deviated from their preregistration. The model-fitting itself is very sophisticated, done in a bayesian manner with stan. Finally, using three different measures of processes (choice, response time, and free sort positioning) is clever.

Having said that, I’m not fully convinced yet by the paper’s conclusions - that people use the SR to identify hierarchy. I follow with some more specific suggestions for potential improvements to the paper:

1. The authors divide up their predictions into preregistered predictions and exploratory analysis. The preregistered predictions are that response times would reflect the SR and choices would reflect a strategy enabled by explicit hierarchical abstraction (EH). It is worth noting that this analysis alone would not really provide support that the SR, as revealed by response times, is connected to the explicit hierarchical abstraction as revealed by choice behavior. The key analysis which connects the response-time SR to choice-time hierarchical abstraction is an exploratory analysis, presented in Fig. 7. Here, it is shown that between-participant variability in the ability of the SR, fit to individual response times, to produce a marker of hierarchical abstaction (“Modularity”), explains between-participant variability in some makers of hierarchical abstraction in both choice and sorting behavior.

I think, in this analysis, the use of the SR “modularity” measure is somewhat obscuring a clear presentation of what precise aspects of variance in response times are relating to makers of hierarchical behavior in choice. My suspicion is that the key variability which SR “modularity” explains is not really specific to the SR itself, but rather mostly reflects variance in whether individuals tend to slow down after they transition from one cluster to another -- a clear marker of hierarchical abstraction in response time behavior -- though one which more models than just the SR could paramaterize. In this sense, I suspect the same variance would also be captured by the variance in the binary regressor in the EH model which tests whether participants changed clusters. Is this the case? If it is, it would not necessarily provide evidence that the SR is not the right model of response times, but it would suggest that the key variance in response times that relates to variance in evidence of hierarchy in choice is not specific to the SR model. More generally, it would permit a potentially simpler story that between participant variance in evidence for hierarchy in choices is related to between participant variance in evidence for hierarchy in response time (as opposed to parameterizing this variance with different models for choice and response time).

2. My other central methodological critique concerns the model comparison for response time data. The paper doesn’t explicitly consider alternatives to the SR for discovering hierarchy. In this sense, the explicit hierarchical (EH) model is a stand-in for potential alternative approaches. Though, I felt like it was not entirely clear what aspects of the RT data led the SR to have an advantage over EH, and I worry that the advantage could potentially be due to superficial choices about what regressors are included for either model.

2a. Specifically, SR and EH don’t seem completely well-matched across types of regressors. As a model of response times, SR has 4 non-nuissance regressors: a) Current choice value, b) last RPE (difference in value between successive states), c) State prediction error (difference in SR between last two successive states), d) Current choice conflict - (difference between choice values for an action)

EH in contrast only has 3 non-nuissance regressors: a) a Binary regressor indicating whether one is in the goal room or not (comparable to SR EV), b) a Binary regressor indicating whether last transition between wings or within wings (comparable to the SR SPE) c) binary regressor indicating whether one is in a room allowing for a transition (somewhat comparable to the SR’s conflict regressor, though potentially not exactly).

This comparison shows that the EH model would be more directly comparable to the SR model if it added regressors comparable to the SR model (but without within-cluster variation) in the following dimensions: RPE and Conflict. Both of these can be derived by considering that the EH model effectively has 3 abstract states (one for each cluster in the task) and assigns a separate Reward for each of those abstract states. This abstract reward is 1 for the goal-containing cluster and 0 for the two non-goal clusters (deriving the ‘value’ of these abstract states is slightly more complicated as it needs to account for most ‘transitions’ leading back to the same state in which the action was taken, but we know that the value of the goal containing state is greater than the value of the two non-goal containing states).

For the RPE component then, for EH this will be 0 following within-cluster transitions, something positive following a transition from a non-goal containing cluster to a goal-containing cluster, and something negative following a transition from a goal-containing cluster to a non-goal-containing cluster.

For the conflict component, this will be 0 when both actions lead to no between-cluster-transisition, one value when one action leads to transitioning between a goal-containing-cluster and a non-goal-containing-cluster, and either a different value or 0 when one action leads to transitioning between two non-goal-containing-cluster. Overall, I’d say, since the conflict component doesn’t seem to significantly capture variance, adding the RPE component is more essential, but something more comparable to conflict would also be desirable so that the different model can be compared in their ability to predict the same type of quantities effect on response time, though they might differ in what they think that quantity is.

2b. More generally, if the EH model is viewed as an RL model that just has 3 abstract states, there should really only be two potential sources of variance that the SR could capture that the EH doesn’t. The first would be variance in response times pertaining to different within-cluster events. EH can’t explain these because it treats the whole cluster as a single state. This would include RTs following within cluster-transitions (for various prediction error regressors) or RTs for conflict for actions that might transition to different within-cluster nodes (for conflict regressors). The second source would have to do with the policy-dependence of the SR - there might be some effect of previous trial’s goals on the SR which in turn would cause variance in response times for upcoming trials. I think showing that the SR captures variance on either of these dimensions would boost confidence that the SR indeed is capturing variance in response times that the EH model can’t (for a version of EH which might be more flexible than the specific one the current paper considers). As an example, this could potentially be done by limiting analysis to within-cluster transitions and showing that the SR SPE for example explains variance for these transitions alone.

3. It would be helpful if the paper could more clearly spell out the theory for how the proposed model of SR underlying abstraction leads to the type of between-participant RT - Choice relationships observed. If I understood, the broad explanation is that all participants learn the SR, but do so using different discount factors. All participants then apply computations to those SRs which enable discovery of abstractions which are used to guide choice, but not response times. However, those computations are only successful for SRs that are defined with high enough discount factors. Thus, only participants whose response times indicate high SR discount factors show evidence of hierarchy in choice. Is this the broad idea? If so, it might be nice to consider what applying secondary computations to SRs that are defined with lower discount factors would look like. Would this produce a recognizable choice behavior distinct from EH? If so, that might provide good additional evidence for this model.

4. I was not convinced really by the explanation given in the discussion for why participants would adopt the EH strategy for choice, given that they use the SR for response times. The discussion argues that the EH strategy is chosen because it is less cognitively demanding and faster to execute. But why then would response times not also adopt and EH strategy? Relatedly, if SR values are computed for response times, why would it be cognitively demanding to then use those already computed values for choices as well?

5. In the discussion, some cited papers are misdescribed. In one example, the discussion points to the following papers as having proposed a mechanism by which the SR can generate hierarchical representations (line 662): McNamee et al., 2016; Tomov et al., 2020; Correa et al., 2022; Stachenfeld et al., 2017. Of these papers, I think only the Stachenfeld paper actually proposes that the SR is used for hierarchical abstraction and proposes a computational process for this. McNamee proposes that abstraction is based on an information theory measure. Tomov proposes that it is based on bayesian inference given specific hierarchical graph priors. The Correa paper is about subgoal selection rather than state abstraction and doesn’t address the SR. I point this out only because I think it would be beneficial for this paper to somewhat more engage with previously made alternative proposals for what might guide state abstraction (and don’t say suggest that the SR is used for it). For example, the Tomov paper in particular presents analysis that explicitly argues against the SR as a model of abstraction. It would be worth it to consider the different conclusions reached there versus here.

Reviewer #3: The authors investigated the computational strategy employed by human participants engaged in a navigation task that is marked by statistical regularities in the form of a “community structure”. The main outcome is that empirically observed response times were best mimicked by a successor representation mdoel, whereas choice behavior was best modeled by a model-based strategy.

I found the question that the authors aim to address of significant interest, and, as far as I could judge, the results convincing. That said, I struggled a lot with this article, and found it hard to comprehend many aspects of the paper. While this could be partly due to a lack of expertise on my part, I think the authors could do a much better job in explaining the question, the logic, the experiment, the results, and their interpretation.

I will provide the main points of confusion below, in the hope that it will be useful for the authors to consider them to improve the clarity of their manuscript.

Task: In the introduction, it is stated that the participant’s choice was probabilistically mapped to two of the four possible neighbouring rooms. But if I understood well, the two options were equiprobable. What was the motivation for these “balanced” action-outcome mappings? Are they required to adjudicate between the different models? Would the different models make similar predictions if the task structure would be more “conventional”, as in eg the original Schapiro study using community structure settings?

Models: All the models include regressors that appear fairly ad-hoc. For example, a regressor modeling the assumed increase in frustration for longer miniblocks, leading to speeded responses. Or a regressor modeling (inverse) “conflict” when there was a difference in “expected value” (with EV quantified how? Distance to goal?). Were all these extra regressors defined a priori, or a posteriori? Do the different models have different amount of explanatory regressors? If so, can that account for their ability to model the observed data? Is this taken into account in any way?

Data: I would have liked to see some more “primary” data, showing eg the average amount of actions that participants needed to get to their goal state; and whether/how this changed over time, as they learnt the community structure. As it stands, Figure 2 immediately jumps to ‘model evidence’ for the different models.

Hypotheses: I found it difficult to understand how the SR model differed from the “optimal model-based decision maker”. It is stated that the SR model holds a multi-step predictive model, whereas the model-based model holds a one-step predictive model. But why is that the case? In MB decision-making, one would also calculate the value of all the future possible actions and use those to derive the optimal action policy? In the cited paper by Momennijad, it was nicely explained how an experimental setting was created that could distinguish between MB and SR. Here, I could not understand how the models would lead to different predictions.

**Have the authors made all data and (if applicable) computational code underlying the findings in their manuscript fully available?**

Reviewer #1: Yes

Reviewer #2: Yes

Reviewer #3: Yes

PLOS authors have the option to publish the peer review history of their article (what does this mean?). If published, this will include your full peer review and any attached files.

Reviewer #1: **Yes: **Christopher W. Lynn

Reviewer #2: No

Reviewer #3: No
---

## [Decision Letter · Decision Letter 1]

5 Feb 2024

Dear Dhr. Wientjes,

We are pleased to inform you that your manuscript 'The successor representation subserves hierarchical abstraction for goal-directed behavior' has been provisionally accepted for publication in PLOS Computational Biology.

Best regards,

Lyle Graham

Section Editor

PLOS Computational Biology

Reviewer's Responses to Questions

**Comments to the Authors:**

Reviewer #1: The authors have adequately addressed my comments and concerns. I believe that the study could have been more effective if the authors used an action-outcome mapping where different actions could not lead to the same outcomes. Fundamentally, this would introduce symmetries between transitions and nodes that would allow the authors to remove some of the regressors in their analysis (which is always desirable). I am tempted to ask the authors to re-run their experiment with transition and node symmetries obeyed, but I don't think their choice of action-outcome map rises to the level of a critical error. I therefore recommend accept.

Reviewer #2: The revisions have satisfied my concerns. I really appreciate the additional models that the authors fit, and the additional analysis looking at whether the SR explains variance beyond transitions between clusters. I also appreciate the new discussion around what secondary computations are ruled in versus out. I encourage the authors to pursue this further in future work.

Reviewer #3: The authors have adequately addressed all the issues raised.

**Have the authors made all data and (if applicable) computational code underlying the findings in their manuscript fully available?**

Reviewer #1: None

Reviewer #2: Yes

Reviewer #3: Yes

PLOS authors have the option to publish the peer review history of their article (what does this mean?). If published, this will include your full peer review and any attached files.

Reviewer #1: **Yes: **Christopher W Lynn

Reviewer #2: No

Reviewer #3: No

---

## [Editor Report · Acceptance letter]

13 Feb 2024

PCOMPBIOL-D-23-01033R1 

The successor representation subserves hierarchical abstraction for goal-directed behavior

Dear Dr Wientjes,

I am pleased to inform you that your manuscript has been formally accepted for publication in PLOS Computational Biology. Your manuscript is now with our production department and you will be notified of the publication date in due course.

With kind regards,

Zsofia Freund
